# Large-scale Uncertainty Quantification for Latent Variable Models Using Subsampling Markov Chain Monte Carlo

**Xiaoyu Wang** [1]   **Jonathan Huggins** [1,2]

## Abstract

Stochastic gradient Langevin dynamics combined with Gibbs updates (SGLD–Gibbs) provides a highly scalable approach to approximate Bayesian inference in latent variable models. However, it remains unclear how to tune the algorithm's hyperparameters in a principled manner to ensure the uncertainty estimates are statistically meaningful. In this work, we address this gap in tuning guidance by developing a statistical scaling limit theory for SGLD–Gibbs. We derive a joint asymptotic limit for the global parameters and latent variables under appropriate space-time rescaling. We show that global parameters converge to a diffusion-type limit, while each latent variable converges to a jump process, reflecting the use of intermittent Gibbs updates. This joint jump-diffusion structure reveals how latent-variable randomness contributes to the stationary distribution of the global parameters. We leverage our results to propose explicit guidance on hyperparameter tuning for SGLD–Gibbs that ensures meaningful uncertainty quantification. Numerical experiments show that SGLD–Gibbs with our tuning guidance leads to better parameter estimates, uncertainty quantification, and predictive performance than stochastic variational inference.

## 1. Introduction

Stochastic gradient methods such as stochastic gradient descent (SGD) and stochastic gradient Langevin dynamics (SGLD) have become central tools for large-scale optimization and approximate Bayesian inference (Nemirovski et al., 2009; Moulines & Bach, 2011; Bottou et al., 2018; Welling & Teh, 2011; Nemeth & Fearnhead, 2021). For approxi-

mate sampling, latent variable models (LVMs) are one of the most frequently cited applications of SG(L)D. Examples include Gaussian mixture models, mixed-membership stochastic block models (Li et al., 2016), latent Dirichlet allocation (Patterson & Teh, 2013), Bayesian matrix factorization (Ahn et al., 2015), mixed effects models (Danaher, 2023), and discrete choice models (Loaiza-Maya & Nibbering, 2023; Loaiza-Maya et al., 2024). In these applications, an *SGLD–Gibbs* scheme is often used, where SGLD update steps are constructed using one or more conditional draws of the latent variables, enabling approximate posterior sampling with per-iteration costs that scale favorably with data size.

However, there is little rigorous guidance on tuning the algorithmic hyperparameters of SGLD–Gibbs such as the step size, minibatch size, and inverse temperature. Moreover, how to obtain meaningful uncertainty quantification using SGLD–Gibbs remains unclear. A substantial body of work (Walk, 1977; Pflug, 1986; Kushner & Yin, 2003; Negrea et al., 2023; Wang et al., 2025) has employed scaling-limit analyses to study standard SG(L)D. These approaches relate SG(L)D sample paths to continuous-time stochastic processes and yield characterizations of optimization accuracy, asymptotic behavior, and numerical efficiency. Such analyses have proven particularly useful for understanding hyperparameter tuning and uncertainty quantification (Mandt et al., 2017; Negrea et al., 2023). However, existing scaling-limit results do not directly apply to latent variable models.

In this work, we address this gap by jointly analyzing the global parameters and latent variables under appropriate space-time rescaling, which provides a unified asymptotic characterization of SGLD–Gibbs dynamics. We show that the global parameters converge to a diffusion-type limit, while each latent variable converges to an independent jump process. We further demonstrate that the interaction between the global diffusion and latent-variable jumps fundamentally alters the noise structure of the global parameters. In particular, the latent variables contribute an additional source of variability determined by the number of Gibbs samples used per iteration. We use our results to derive concrete guidance for uncertainty quantification and hyperparameter tuning.

---

[1]Department of Mathematics & Statistics, Boston University [2]Faculty of Computing & Data Sciences, Boston University. Correspondence to: Xiaoyu Wang <shawnwxy@bu.edu>.

*Proceedings of the 43rd International Conference on Machine Learning*, Seoul, South Korea. PMLR 306, 2026. Copyright 2026 by the author(s).

Empirically, we find that SGLD–Gibbs yields improved accuracy and more reliable uncertainty quantification compared to stochastic variational inference in applications to mixture modeling and topic modeling.

## 1.1. Related Work and Alternative Approaches

Given their widespread use, SG(L)D methods have been studied from many perspectives, including finite-sample error bounds, convergence rates, and stationary distributions (e.g., Mcleish, 1976; Ruppert, 1988; Polyak & Juditsky, 1992; Kushner & Yin, 2003; Negrea et al., 2023; Rakhlin et al., 2011; Dieuleveut et al., 2020; Mou et al., 2020; Cheng et al., 2020; Srikant, 2024; Anastasiou et al., 2019; Ge et al., 2015; Jin et al., 2017). Most relevant to our work is scaling-limit theory for stochastic approximation algorithms, which shows that, under appropriate rescaling, SGD and SGLD trajectories converge to Ornstein–Uhlenbeck diffusions (Kushner & Huang, 1981; Kushner & Yang, 1993; Kushner & Yin, 2003; Negrea et al., 2023). Further results characterize mixing times, stationary covariances, and the behavior of averaged iterates (Mandt et al., 2017; Negrea et al., 2023; Collins-Woodfin et al., 2024; Qian et al., 2024; Kushner & Yang, 1993).

Several recent works extend this theory to improve uncertainty quantification for stochastic gradient algorithms. Wang et al. (2025) develop non-asymptotic functional error bounds for diffusion approximations to scaling limits. Wang et al. (2026) develop discrete-time proxy theories for stochastic gradient algorithms, clarifying when diffusion-based uncertainty quantification remains valid in large-batch or non-asymptotic regimes. A separate line of work studies scaling limits of SGD in high-dimensional regimes where the parameter dimension $d \to \infty$, yielding mean-field or dynamical equations for low-dimensional summary statistics (Arous et al., 2022; Collins-Woodfin et al., 2023; Mignacco et al., 2021).

Variational Bayesian methods, including mean-field variational Bayes, online variational Bayes, stochastic variational inference (SVI), and related variational approximations, can also provide scalable for latent variable model inference (Hoffman et al., 2013; 2010; Kucukelbir et al., 2017). Nonetheless, their ability to quantify posterior uncertainty can be fundamentally limited (Gelman et al., 2013; Margossian et al., 2025; Giordano et al., 2018). For example, Margossian et al. (2025) show that when the true posterior distribution exhibits dependence structure, variational approximations based on factorization cannot, in general, correctly estimate posterior uncertainty. Depending on the divergence being minimized, uncertainty estimates produced by variational Bayes are often poorly calibrated, even under correct model specification.

## 2. Preliminaries and Problem Setup

This section introduces the class of latent variable models considered in this work, describes the SGLD–Gibbs algorithm, and reviews preliminary results concerning scaling limits for stochastic gradient methods.

### 2.1. Latent Variable Models

We consider a general class of latent variable models in which each observation is associated with an unobserved latent variable. Let $\{(X_i, z_i)\}_{i=1}^n$ denote independent pairs of observed data $X_i \in \mathcal{X}$ and latent variables $z_i \in \mathcal{Z}$. The joint distribution is parameterized by a global parameter $\theta \in \Theta \subset \mathbb{R}^d$ and admits the factorization

$$p(X_i, z_i \mid \theta) = p(z_i \mid \theta)\, p(X_i \mid z_i, \theta),$$

with prior distribution $\pi_0(\theta)$ on $\theta$. The marginal likelihood of the observations is given by $p(X_i \mid \theta) = \int p(X_i, z_i \mid \theta)\, dz_i$, and the corresponding log-likelihood is $\ell(\theta; X_i) := \log p(X_i \mid \theta)$. This formulation encompasses a wide range of commonly used models, including mixture models, mixed-membership stochastic block models, topic models, and Bayesian matrix factorization. See Murphy (2023) for a systematic discussion of learning and approximate Bayesian inference in such models.

### 2.2. SGLD with Gibbs Updates

We study stochastic gradient Langevin dynamics combined with Gibbs updates for latent variables. Let $b \in \{1, \ldots, n\}$ denote the minibatch size. At iteration $k$, a minibatch of indices $I_k = \{I_k(1), \ldots, I_k(b)\}$ is sampled uniformly from $\{1, \ldots, n\}$ with replacement. This convention is mainly for theoretical convenience. Similar scaling-limit results for sampling without replacement were established in Negrea et al. (2023), and we expect the conclusions here to remain the same under sampling without replacement, up to constant-level deviations when $b$ is of order $n$. Given the current global parameter $\theta_k$, the algorithm proceeds in two steps:

**(i) Gibbs updates of latent variables.** For each $i \in I_k$, the latent variable is resampled from its conditional posterior,

$$z_{i,k+1} \sim p(z_i \mid X_i, \theta_k),$$

while latent variables not in the minibatch remain unchanged.

**(ii) SGLD update of global parameters.** Using the stochastic gradient estimator with refreshed latent variables

**Algorithm 1** SGLD–Gibbs for Latent Variable Models
___
1: **Input:** step size $h$, batch size $b$, inverse temperature $\beta$, preconditioner $\Gamma$, initial values $(\theta_0, \{z_{i,0}\}_{i=1}^n)$
2: **for** $k = 0, 1, 2, \ldots$ **do**
3:   Sample minibatch $I_k \subset \{1, \ldots, n\}$ with $|I_k| = b$
4:   **for** each $i \in I_k$ **do**
5:     Sample $z_{i,k+1} \sim p(z_i \mid X_i, \theta_k)$
6:   **end for**
7:   Update $\theta_{k+1}$ using the SGLD step with updated $\{z_{i,k+1}\}_{i \in I_k}$
8: **end for**
___

given by

$$
\begin{aligned}
G_k(\theta) := \ &\frac{1}{n} \nabla \log \pi_0(\theta) \\
&+ \frac{1}{b} \sum_{i \in I_k} \nabla_\theta \log p(X_i, z_{i,k+1} \mid \theta),
\end{aligned}
\tag{1}
$$

the global parameter is updated via

$$
\theta_{k+1} = \theta_k + \frac{h}{2} \Gamma\, G_k(\theta_k) + \sqrt{\frac{h}{\beta}}\, \Gamma^{1/2} \xi_k,
\tag{2}
$$

where $h > 0$ is the step size, $\beta \in (0, \infty]$ is the inverse temperature, $\Gamma \in \mathbb{R}^{d \times d}$ is a positive definite preconditioning matrix, and $\xi_k \sim \mathcal{N}(0, I_d)$. The full procedure is summarized in Algorithm 1.

### 2.3. Scaling Limits for Stochastic Gradient Methods

In a general setup that does not involve latent variables, assume observations are i.i.d. from an unknown distribution $P_\star$. The model is parameterized by a global parameter $\theta \in \Theta \subset \mathbb{R}^d$ and admits a likelihood of the form $p(X_i \mid \theta)$ for each observation $X_i$, with prior distribution $\pi_0(\theta)$ on $\theta$. The optimal parameter is given by $\theta_\star := \arg\min_\theta \mathbb{E}[\ell(X, \theta)]$, where $X \sim P_\star$.

Recall that $I_k \subset \{1, \ldots, n\}$ denotes the $k$th minibatch. SGLD uses the one-step update given in Equation (2), where the stochastic gradient estimator is now

$$
G_k(\theta) := \frac{1}{n} \nabla \log \pi_0(\theta) + \frac{1}{b} \sum_{i \in I_k} \nabla_\theta \log p(X_i \mid \theta).
$$

Scaling limit theory relates discrete-time stochastic gradient algorithms to continuous-time stochastic processes under appropriate space-time rescaling. Let $\theta_k^{(n)} \in \mathbb{R}^d$ denote the global parameter at iteration $k$ and $\hat{\theta}^{(n)}$ denote a critical point satisfying the first-order condition $\sum_{i=1}^n \nabla \ell(\hat{\theta}^{(n)}; X_i) = 0$. Define the rescaled, continuous-time process

$$
\vartheta_t^{(n)} = n^{\mathfrak{w}} \left( \theta_{\lfloor n^{\mathfrak{a}} t \rfloor}^{(n)} - \hat{\theta}^{(n)} \right),
\tag{3}
$$

where $\mathfrak{w} > 0$ and $\mathfrak{a} > 0$ denote the spatial and temporal scaling exponents.

Then, under an appropriate scaling regime, this process converges in distribution to an Ornstein–Uhlenbeck process whose drift and diffusion coefficients depend on the preconditioner $\Gamma$ and the first- and second-order Fisher information matrices

$$
I_\star := \mathbb{E}\left[ [\nabla_\theta \ell(\theta^\star; X)]^{\otimes 2} \right], \quad J_\star := -\mathbb{E}\left[ \nabla_\theta^{\otimes 2} \ell(\theta^\star; X) \right],
$$

where $a^{\otimes 2} := a \otimes a$ denotes the outer product, and $\nabla^{\otimes 2}$ denotes the Hessian operator.

Here $I_\star$ quantifies the variability of the log-likelihood gradient at $\theta^\star$, while $J_\star$ captures the local second-order behavior of the log-likelihood around $\theta^\star$.

**Theorem 2.1** (Negrea et al. (2023), Theorem 1). *Consider the SGLD algorithm with step size $h^{(n)} = c_h n^{-\mathfrak{h}}$, batch size $b^{(n)} = \lfloor c_b n^{\mathfrak{b}} \rfloor$, and inverse temperature $\beta^{(n)} = c_\beta n^{\mathfrak{t}}$. Let $\mathfrak{a} = \mathfrak{h}$ and $\mathfrak{w} = \min\{\mathfrak{b} + \mathfrak{h}, \mathfrak{t}\}/2$. Then, under mild regularity conditions, as $n \to \infty$,*

$$
\vartheta_t^{(n)} \Rightarrow \vartheta_t
$$

*in the Skorohod topology in probability, where $\Rightarrow$ denotes weak convergence and $\vartheta_t$ is an Ornstein–Uhlenbeck process solving*

$$
d\vartheta_t = -\frac{1}{2} B \vartheta_t\, dt + \sqrt{A}\, dW_t,
\tag{4}
$$

*with*

$$
\begin{aligned}
B &= c_h \Gamma J_\star \mathbf{1}\{\mathfrak{a} = \mathfrak{h}\}, \\
A &= \frac{c_h}{c_\beta} \Gamma \mathbf{1}\{\mathfrak{h} + \mathfrak{b} \leq \mathfrak{t}\} + \frac{c_h^2}{4 c_b} \Gamma I_\star \Gamma^\top \mathbf{1}\{\mathfrak{t} \leq \mathfrak{b} + \mathfrak{h}\}.
\end{aligned}
$$

As discussed by Negrea et al. (2023), one implication of this result is that characterization of full-path limiting dynamics relates the mixing behavior of SGLD to the spectral properties of the limiting Ornstein–Uhlenbeck process. Heuristically, the asymptotic mixing time then scales inversely to the smallest eigenvalue of the corresponding drift matrix $B$. This motivates choosing the preconditioner $\Gamma$ to approximate $J_\star^{-1}$, which optimizes mixing speed in the asymptotic regime.

### 2.4. Uncertainty Quantification

Theorem 2.1 shows how the scaling of the step size, minibatch size, and inverse temperature determines which noise sources remain active in the limiting diffusion, which determines the form of the stationary covariance, as summarized by the following standard result.

**Proposition 2.2** (Negrea et al. (2023), Corollary 1). *For the Ornstein–Uhlenbeck process $(\vartheta_t)_{t \geq 0}$ defined by (4) with*

| Target | Asymp. cov. | $\Gamma$ | $\beta$ | $h$ |
|--------|-------------|----------|---------|-----|
| BvM | $J_\star^{-1}$ | $I_\star^{-1}$ | $\frac{n}{1-w_1}$ | $\frac{4w_1 b}{n}$ |
| Bagged post. | $w_2 J_\star^{-1} + w_1 S_\star$ | $J_\star^{-1}$ | $\frac{n}{w_2}$ | $\frac{4w_1 b}{n}$ |

*Table 1.* Recommended tuning parameter combinations and the corresponding asymptotic covariances of SGLD in Negrea et al. (2023). **BvM:** targeting the asymptotic covariance of the posterior based on the Bernstein–von Mises. **Bagged post.:** targeting the asymptotic covariance of the (generalized) bagged posterior (Huggins & Miller, 2024). If targeting the sandwich covariance, use the bagged posterior tuning with $w_1 = 1$ and $w_2 = 0$.

*known $\vartheta_0$, the law of $\vartheta_t$ is Gaussian with mean $e^{-tB/2}\vartheta_0$ and covariance*

$$\Sigma_t = \int_0^t e^{-sB/2} A e^{-sB^\top/2}\, ds.$$

*If a stationary distribution exists, then $\vartheta_t$ admits $\mathcal{N}(0, \Sigma_\infty)$ as its stationary law, where $\Sigma_\infty$ solves*

$$\tfrac{1}{2} B \Sigma_\infty + \tfrac{1}{2} \Sigma_\infty B^\top = A.$$

Thus, Proposition 2.2 enables the targeting of a desired stationary covariance to ensure meaningful uncertainty quantification. In the Bayesian setting, the Bernstein-von Mises theorem states that the posterior is approximately $\mathcal{N}(\hat{\theta}^{(n)}, J_\star^{-1}/N)$ (Kleijn & van der Vaart, 2012). Thus, one possible goal when using SGLD for uncertainty quantification is to obtain samples with a distribution that is approximately equal to $\mathcal{N}(\hat{\theta}^{(n)}, J_\star^{-1}/N)$. However, the sampling distribution of $\hat{\theta}^{(n)}$ is asymptotically normal with mean $\theta_\star$ and covariance equal to $J_\star^{-1} I_\star J_\star^{-1}/N$ (White, 1982). The matrix $S_\star := J_\star^{-1} I_\star J_\star^{-1}$ is known as the "sandwich" covariance matrix, and it suggests that for proper uncertainty quantification we want the stationary SGLD distribution to be approximately $\mathcal{N}(\hat{\theta}^{(n)}, J_\star^{-1} I_\star J_\star^{-1}/N)$. When the model is correctly specified, $I_\star = J_\star$, so the sandwich covariance is equal to $J_\star^{-1}$ and the Bayesian posterior provides correct uncertainty quantification. In the misspecified setting, tuning SGLD to have stationary covariance $S_\star/N$ will correctly capture the sampling uncertainty. Alternatively, the model-based and sampling-based uncertainties can be combined, as in the bagged posterior (Huggins & Miller, 2024). Based on these statistical considerations, Negrea et al. (2023) propose to target either the Bernstein–von Mises uncertainty or the bagged posterior uncertainty, with the sampling uncertainty arising as a special case of the latter. The recommended tunings derived from Theorem 2.1 and Proposition 2.2 are summarized in Table 1.

## 3. Main Results

In this section we present our main theoretical results. We first define the scaling-limit objects for both the global parameter and latent variables, together with the associated

*latent-involved Fisher information matrix.* We then state our main joint scaling-limit theorem for SGLD–Gibbs, followed by corollaries on uncertainty quantification and latent-variable mixing. All technical assumptions and proofs are deferred to the supplementary materials.

### 3.1. Scaling-limit objects and information matrices

We take the same definition of rescaled global-parameter process $\vartheta_t^{(n)}$ as in (3) and define latent-variable process by

$$\zeta_{i,t}^{(n)} = z_{i,\lfloor n^{\mathfrak{a}} t\rfloor}^{(n)},$$

where $z_{i,k}^{(n)}$ denote the latent variable associated with observation $i$ at $k$th iteration.

Distinguished from $I_\star$ defined in Section 2.3, we define *latent-involved Fisher information matrix*

$$\widetilde{I}_\star := \mathbb{E}_{X,Z|\theta^\star}\left[\nabla_\theta \log p(X, Z \mid \theta^\star)^{\otimes 2}\right] = I_\star + M_\star,$$

where

$$M_\star := \mathbb{E}_{X|\theta^\star}\left[\mathrm{Var}_{Z|X,\theta^\star}\left(\nabla_\theta \log p(X, Z \mid \theta^\star)\right)\right] \succeq 0$$

is the "Jensen gap" that quantifies the additional *algorithm-induced uncertainty* due to estimating the marginal likelihood with only a single Gibbs sample. It follows from the definition of $M_\star$ that the gap will be smaller when, on average, there is less uncertainty about $Z$ given $X$.

### 3.2. Joint scaling limit for global and latent parameters

We now show that, under an appropriate asymptotic scaling, the rescaled global parameter and a fixed latent variable (e.g., $z_1$) converge jointly in distribution. The limiting process has independent diffusion and jump components. Specifically, the global parameter process converges to an Ornstein–Uhlenbeck process whose drift and diffusion structure is analogous to that in Theorem 2.1, except that the diffusion matrix is modified to explicitly incorporate algorithm-induced uncertainty arising in SGLD–Gibbs. Meanwhile, the latent-variable process converges to a Poisson-driven Gibbs jump process, with jumps drawn from the true conditional posterior of the latent variable.

**Theorem 3.1** (Joint scaling limit of SGLD–Gibbs). *Consider the SGLD–Gibbs algorithm with the same polynomial scaling of tuning parameters as in Theorem 2.1. Let $\mathfrak{a} = \mathfrak{h}$ and $\mathfrak{w} = \min\{\mathfrak{b}+\mathfrak{h}, \mathfrak{t}\}/2$. Assume the regularity conditions stated in Section A.1. Then, as $n \to \infty$,*

$$\left(\vartheta_t^{(n)}, \zeta_{1,t}^{(n)}\right) \Rightarrow \left(\vartheta_t, \zeta_{1,t}\right),$$

*in the Skorokhod topology in probability, where the limiting processes $\vartheta_t$ and $\zeta_{1,t}$ are independent and defined as follows:*

1. The global-parameter limit $\vartheta_t$ is an Ornstein–Uhlenbeck process solving

$$d\vartheta_t = -\frac{1}{2}B\vartheta_t\,dt + \sqrt{A}\,dW_t,$$

   with

$$B = c_h\Gamma J_\star\mathbf{1}\{\mathfrak{a} = \mathfrak{h}\},$$

$$A = \frac{c_h}{c_\beta}\Gamma\mathbf{1}\{\mathfrak{h} + \mathfrak{b} \leq \mathfrak{t}\} + \frac{c_h^2}{4c_b}\Gamma\tilde{I}_\star\Gamma^\top\mathbf{1}\{\mathfrak{t} \leq \mathfrak{b} + \mathfrak{h}\}.$$

2. When $\mathfrak{h} + \mathfrak{b} \leq 1$, the latent-variable limit $\zeta_{1,t}$ is a pure-jump Markov process with generator

$$(\mathcal{L}f)(z) = \lambda\int\left\{f(z') - f(z)\right\}p(z' \mid X_1, \theta^\star)\,dz',$$

   where $\lambda := c_b\mathbf{1}\{\mathfrak{h} + \mathfrak{b} = 1\}$.

For global parameters, the joint scaling limit in Theorem 3.1 yields the same scaling regimes in $(\mathfrak{h}, \mathfrak{b}, \mathfrak{t})$ as Theorem 2.1. The key difference in latent variable models is that the diffusion term involves $\tilde{I}_\star$, which captures additional uncertainty arising from the latent variables and Gibbs sampling. This latent-induced contribution persists in the scaling limit and affects the stationary covariance of the global parameters, beyond what is implied by the marginal likelihood alone.

The behavior of the latent-variable dynamics depends critically on the relation between the scaling exponents $\mathfrak{h}$ and $\mathfrak{b}$. If $\mathfrak{h} + \mathfrak{b} = 1$, each latent variable evolves as a pure-jump Markov process with a nondegenerate limiting intensity, yielding a meaningful joint jump-diffusion limit as characterized in Theorem 3.1. If $\mathfrak{h} + \mathfrak{b} < 1$, the latent-variable dynamics become degenerate on the macroscopic timescale; this is because $z_1$ is refreshed too infrequently relative to the evolution of $\theta$, so the latent-variable process effectively freezes and loses any meaningful limiting behavior. If $\mathfrak{h} + \mathfrak{b} > 1$, latent-variable updates occur on a much faster time scale than the global parameters. In this scenario, the latent-variable generator diverges in the limit, and thus the sequence of latent-variable processes does not admit a well-defined limit.

**Corollary 3.2** (Latent-variable limit: stationarity and mixing). *Assume the regime of Theorem 3.1 with $\mathfrak{h} + \mathfrak{b} = 1$, in which the latent-variable limit is a pure-jump Markov process with rate $\lambda = c_b$. Let $\mu_t$ denote the law of $\zeta_{1,t}$ with initial law $\mu_0$. Define the $\varepsilon$-mixing time in total variation by*

$$t_{\mathrm{mix}}(\varepsilon) := \inf\left\{t \geq 0 : d_{\mathrm{TV}}(\mu_t, \pi) \leq \varepsilon\right\}.$$

*Then $\pi(\cdot) = p(\cdot \mid X_1, \theta^\star)$ is the unique stationary distribution and, for all $t \geq 0$,*

$$\mu_t = e^{-\lambda t}\mu_0 + (1 - e^{-\lambda t})\pi.$$

*Consequently, $d_{\mathrm{TV}}(\mu_t, \pi) = e^{-\lambda t}d_{\mathrm{TV}}(\mu_0, \pi)$, and hence*

$$t_{\mathrm{mix}}(\varepsilon) = \frac{1}{\lambda}\log\left(\frac{1}{\varepsilon}\right).$$

This result implies that $\zeta_{1,t}$ evolves as a refresh process that periodically resamples from $p(z_1 \mid X_1, \theta^\star)$. Since each latent-variable update acts as an instantaneous refresh in the scaling limit, the effective mixing time is of order $1/\lambda$. For the global parameter, the relevant mixing heuristic is inherited from Theorem 2.1 via the drift matrix $B$ of the limiting Ornstein–Uhlenbeck process, as in Negrea et al. (2023)

### 3.3. Scaling limit under generalized Gibbs approximation

We now consider a natural extension of the SGLD–Gibbs update that is commonly used in practice, in which the single latent-variable draw in the stochastic gradient is replaced by an average over multiple conditional samples. Letting $S$ denote the number of Gibbs samples per iteration, for each $i \in I_k$ now draw $z_{i,k+1}^{(1)}, \ldots, z_{i,k+1}^{(S)} \stackrel{\text{i.i.d.}}{\sim} p(z_i \mid X_i, \theta_k)$. We then define the averaged stochastic gradient

$$G_k^{(S)}(\theta) := \frac{1}{n}\nabla\log\pi_0(\theta)$$
$$+ \frac{1}{bS}\sum_{i \in I_k}\sum_{s=1}^{S}\nabla_\theta\log p\left(X_i, z_{i,k+1}^{(s)} \mid \theta\right).$$

For $S = 1$, this reduces to the standard SGLD–Gibbs gradient $G_k(\theta)$ defined in Equation (1). Theorem 3.1 generalizes to this case, with the only difference being that the diffusion matrix $A$ is replaced by

$$A_S = \frac{c_h}{c_\beta}\Gamma\mathbf{1}\{\mathfrak{h} + \mathfrak{b} \leq \mathfrak{t}\}$$
$$+ \frac{c_h^2}{4c_b}\Gamma\widetilde{I}_\star^{(S)}\Gamma^\top\mathbf{1}\{\mathfrak{t} \leq \mathfrak{b} + \mathfrak{h}\},$$

where $I_\star^{(S)} := I_\star + \frac{1}{S}M_\star$. Hence, we see that when $S > 1$, the "Jensen gap" $M_\star$ decreases to $M_\star/S$ and goes to zero as $S \to \infty$ (i.e., as the marginal likelihood estimate error goes to zero). However, smaller algorithm-induced uncertainty comes at the cost of $S$ times as many gradient evaluations per iteration. Our result thus helps quantify the trade-off between computational cost and accuracy of the uncertainty quantification.

### 3.4. Parameter Tuning and uncertainty quantification

To retain a nondegenerate and interpretable joint scaling limit, we focus on the regime $\mathfrak{h} + \mathfrak{b} = 1$, in which the latent-variable component admits a nontrivial macroscopic limit. Our discussion covers general $S \geq 1$, with the special case

| Target | Asymp. cov. | $\Gamma$ | $\beta$ | $h$ |
|---|---|---|---|---|
| Bernstein–von Mises | $J_\star^{-1}$ | $\big(\widetilde{I}_\star^{(S)}\big)^{-1}$ | $n/(1-w_1)$ | $4w_1 b/n$ |
| Bagged post. {+ algorithmic variability} | $w_1 S_\star + w_2 J_\star^{-1} + \dfrac{w_1}{S} J_\star^{-1} M_\star J_\star^{-1}$ | $J_\star^{-1}$ | $n/w_2$ | $4w_1 b/n$ |

*Table 2.* Recommended tuning parameter combinations and the corresponding asymptotic covariances of SGLD–Gibbs. See the caption of Table 1 for further explanation.

$S = 1$ recovering the standard SGLD–Gibbs update with a single conditional draw. To keep the Gaussian noise active in the limit, we also set $\mathfrak{t} = 1$. Thus, we parameterize the tuning constants following the SGLD scaling-limit literature by setting

$$\beta^{(n)} = \frac{n}{w_2}, \quad \text{and} \quad h^{(n)} = \frac{4w_1 \, b^{(n)}}{n},$$

where $w_1, w_2 > 0$ are constants. With these choices, the stationary covariance of the limiting OU process depends on the choice of preconditioner $\Gamma$ and on $S$. This leads to two natural tuning strategies.

**Bernstein–von Mises covariance via $\Gamma = \big(\widetilde{I}_\star^{(S)}\big)^{-1}$.** If we choose the preconditioner $\Gamma = \big(\widetilde{I}_\star^{(S)}\big)^{-1}$, then, by taking the weights to satisfy $w_1 + w_2 = 1$, the stationary covariance is $J_\star^{-1}$, recovering Bayes-type uncertainty quantification. Note that taking $w_1 = 0$ results in using SGD–Gibbs rather than SGLD–Gibbs.

**Bagged posterior and sandwich-type covariances via $\Gamma = J_\star^{-1}$.** Alternatively, if we choose $\Gamma = J_\star^{-1}$, the stationary covariance takes the bagged posterior form

$$w_1 \, J_\star^{-1} \, \widetilde{I}_\star^{(S)} \, J_\star^{-1} + w_2 \, J_\star^{-1}$$
$$= w_1 J_\star^{-1} I_\star J_\star^{-1} + w_2 J_\star^{-1} + \frac{w_1}{S} J_\star^{-1} M_\star J_\star^{-1}.$$

The third term isolates the algorithm-induced contribution and decays as $S$ increases. A sandwich-type covariance can thus be recovered by taking $w_1 = 1$ and $w_2 = 0$. The resulting tuning recommendations are summarized in Table 2.

### 3.5. Proof sketch of Theorem 3.1

Our proof follows the spirit of Negrea et al. (2023). We establish weak convergence of the processes in the Skorokhod topology in probability by first proving almost sure convergence along subsequences. This is achieved by showing that the difference between the approximate generator and the limiting generator, evaluated on smooth test functions with compact support, vanishes uniformly. We divide the proof into two parts.

In Part 1, we consider arguments that are sufficiently far from the support of the test function. The main idea is to control the probability that the global-parameter process jumps back into the support of the test function. A key difference with Negrea et al. (2023) is that we must impose assumptions on the joint likelihood uniformly on latent variable values to ensure that, even in the presence of additional uncertainty induced by the latent-variable updates, the probability that the global parameter jumps back into the support can still be controlled at a comparable scale.

In Part 2, we analyze arguments that lie in or are close to the support of the test function. We perform a Taylor expansion of the joint approximate generator. The drift term converges in the same way as in Negrea et al. (2023). For the gradient component of the diffusion term, a similar approach applies, but the resulting limit now explicitly incorporates additional variability arising from sampling the latent variables. We then have to control a new term that bridges the infinitesimal operator associated with Gibbs updates and the generator of a Poisson jump process. The most technically challenging new aspect is to show that all cross terms involving both the global parameters and the latent variables vanish in the limit. This vanishing further implies that, in the asymptotic regime, the contribution of any single observation becomes asymptotically independent of the global-parameter dynamics, which in turn allows the joint limiting distribution to factorize into independent components.

## 4. Experiments

Our experiments are designed to answer three questions. First, we verify the predictions of our scaling-limit theory: under the hyperparameter tuning guided by Table 2, samples produced by SGLD–Gibbs should match the stationary distributions predicted by the limiting processes. On synthetic models with known ground-truth parameters, we directly compare empirical posteriors of both global parameters and latent variables with their theoretical stationary distributions. Second, we evaluate the quality of uncertainty quantification provided by SGLD–Gibbs and compare it with stochastic variational inference (SVI), which we assess using posterior variances and rank-uniformity calibration diagnostics. Third, on real datasets where direct verification against ground-truth parameters is not possible, we evaluate downstream performance using clustering accuracy (ARI and AMI) for mixture models and held-out perplexity for topic models. Complete experimental details are provided

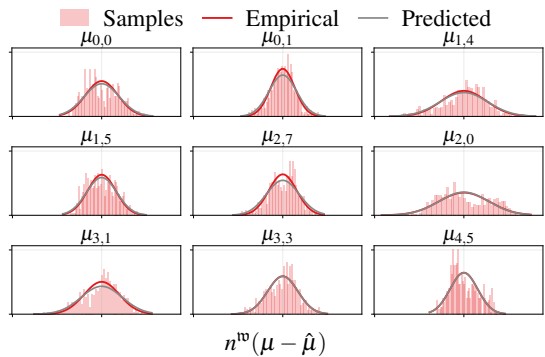

**a.** Empirical stationary density of the rescaled global parameter $\vartheta^{(n)}$ under SGLD–Gibbs, compared with the Ornstein–Uhlenbeck stationary distribution predicted by the scaling limit.

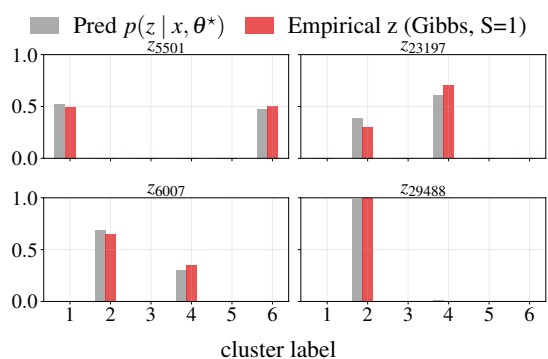

**b.** Empirical stationary behavior of representative latent assignments, consistent with the jump-type limiting dynamics.

*Figure 1.* Validation of the joint scaling limit on a synthetic Gaussian mixture model.

in Appendix C.

### 4.1. Synthetic Gaussian Mixture Model

We generate synthetic data with sample size $n = 30,000$ with observations $x_i \in \mathbb{R}^8$. The data are drawn from a finite Gaussian mixture with 6 clusters. We run SGLD–Gibbs using a minibatch size $b = 50$ and consider Gibbs updates with $S \in \{1, 10\}$ samples. For SGLD, we follow the sandwich tuning and choose the step size and inverse temperature as $h = 2b/n$ and $\beta = 2n$. We use a preconditioner for the global parameters constructed from an estimate of $J_\star^{-1}$, as described in Table 2. As a baseline, we apply SVI with a standard mean-field variational family. We use a diagonal-covariance GMM with Normal–Gamma variational factors for the component means and precisions.

Figure 1 demonstrates that our scaling limit theory accurately predicts the distributions of the global and latent variables. As shown in Figure 2(a), SGLD–Gibbs also produces substantially better calibrated uncertainty than SVI, with empirical ranks lying close to the uniform reference. Finally, Figure 2(b) demonstrates that SGLD–Gibbs also provides more accurate parameter estimates. Moreover, as predicted by our theory, increasing the number of Gibbs draws to $S = 10$ leads to a clear reduction in posterior uncertainty, resulting in visibly tighter marginal distributions compared to the $S = 1$ case.

### 4.2. Synthetic Latent Dirichlet Allocation

We generate a synthetic topic model with vocabulary size $V = 50$, number of topics $K = 3$, and corpus size $d = 10000$ documents. Each document is generated from a Latent Dirichlet Allocation (LDA) model with fixed topic proportions and topic-word distributions. We run SGLD–Gibbs using a minibatch size $b = 100$ and consider Gibbs updates with $S \in \{1, 5\}$ samples. For SGLD we use the

bagged posterior tuning with $w_1 = w_2 = 1$ (so $h = 4b/n$ and $\beta = n$) and for the preconditioner we follow (Patterson & Teh, 2013). This preconditioner is motivated by the natural geometry of the simplex, taking a diagonal form proportional to $\mathrm{diag}(\theta)$. The resulting algorithm is known as *stochastic gradient Riemannian Langevin dynamics* (SGRLD). As a baseline, we apply stochastic variational inference (SVI) to a semi-collapsed LDA model. The document-level topic proportions are integrated out analytically, and we use a mean-field variational family in which the global topic-word distributions are modeled by Dirichlet factors and the latent topic assignments by categorical factors.

Figure 3(a) shows that SGRLD–Gibbs yields ranks substantially closer to uniform than SVI, indicating superior calibration. Figure 3(b) shows that the SGRLD–Gibbs posterior concentrates more tightly around the ground-truth values and better matches the target uncertainty, whereas SVI exhibits noticeable miscalibration and bias in several coordinates. Using $S = 5$ only slightly reduces uncertainty compared to $S = 1$ (far less than in GMM), likely due to the small $S$ and the strong autocorrelation of Gibbs updates in LDA.

### 4.3. Real-data evaluation: Flow Cytometry (GMM) and 20 Newsgroups (LDA)

We further evaluate SGLD–Gibbs on two real-world latent variable models: a diagonal-covariance Gaussian mixture model (GMM) on a flow cytometry dataset and latent Dirichlet allocation (LDA) on the 20 Newsgroups corpus.

**Flow Cytometry (GMM).** We measure inferential quality in terms of the adjusted Rand index (ARI) and adjusted mutual information (AMI), which compares the inferred clustering to the provided expert clustering. Table 3 shows

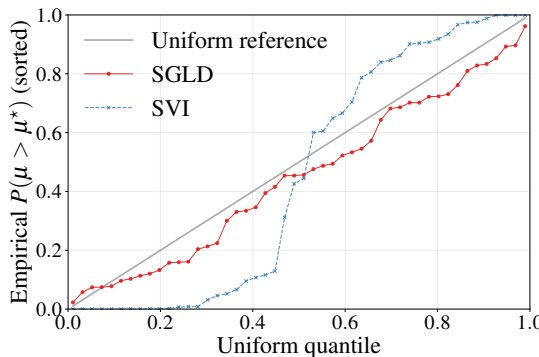

**a.** Rank-uniformity diagnostic for global parameters. SGLD–Gibbs yields empirical ranks closer to the uniform reference line than stochastic variational inference (SVI), indicating better-calibrated uncertainty.

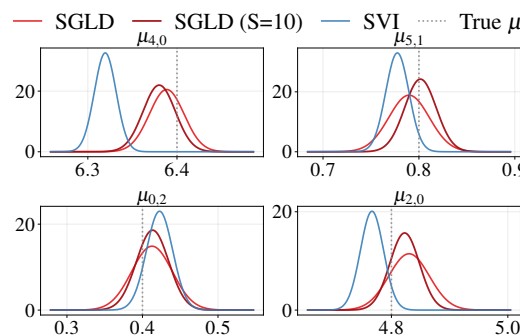

**b.** Representative posterior marginals for selected parameters under SGLD–Gibbs and SVI, together with the true parameter values. SGLD–Gibbs concentrates more accurately around the ground truth. Increasing $S$ reduces uncertainty significantly.

*Figure 2.* Synthetic Gaussian GMM: uncertainty quantification and posterior accuracy

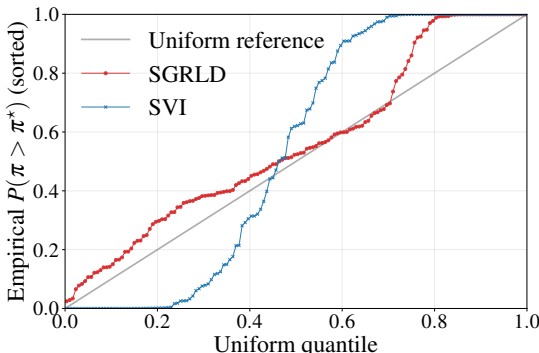

**a.** Rank-uniformity diagnostic over topic-word probabilities. SGRLD–Gibbs yields empirical ranks closer to the uniform reference line, indicating better-calibrated uncertainty.

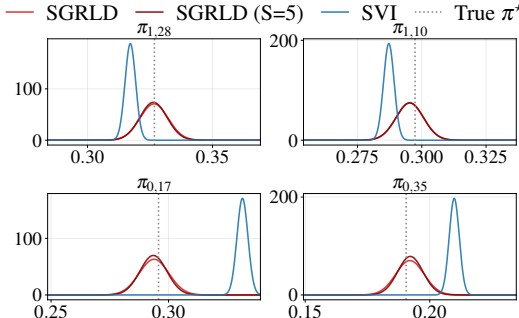

**b.** Posterior marginals for selected topic-word probabilities under SGRLD–Gibbs and SVI, with true parameter values indicated. SGRLD–Gibbs concentrates more accurately around the ground truth with higher uncertainty. Increasing $S$ slightly reduces uncertainty.

*Figure 3.* Synthetic LDA: uncertainty calibration and posterior accuracy.

*Table 3.* Performance comparison on real datasets. For Flow Cytometry (GMM), clustering quality is evaluated using adjusted Rand index (ARI) and adjusted mutual information (AMI). For 20 Newsgroups (LDA), predictive performance is measured by held-out perplexity (lower is better).

| DATASET | METHOD | METRIC | RESULT |
|---|---|---|---|
| FLOW CYTOMETRY | SVI | ARI / AMI | 0.47 / 0.68 |
| | SGLD–GIBBS | ARI / AMI | **0.80** / **0.82** |
| 20 NEWSGROUPS | SVI | PERPLEXITY | 2402 |
| | SGRLD–GIBBS | PERPLEXITY | **2052** |

that SGLD–Gibbs attains substantially higher ARI and AMI than SVI.

**20 Newsgroups (LDA).** For LDA, we compare methods based on predictive performance using held-out perplexity. As shown in Table 3, SGLD–Gibbs achieves much better perplexity ($\sim 350$ nats) than SVI.

Overall, these real-data results complement our synthetic experiments, demonstrating that in practice SGLD–Gibbs retains the scalability of SVI but provides superior task-specific performance.

# 5. Discussion and Future Work

This work provides a scaling-limit perspective on SGLD–Gibbs in latent variable models, clarifying how uncertainty quantification and algorithmic tuning are shaped by the interaction between stochastic-gradient dynamics and latent-variable updates. The resulting joint jump–diffusion limit explains how additional algorithm-induced uncertainty due to estimating the marginal likelihood with Gibbs samples contributes to the effective noise of the global parameters and yields principled guidance for hyperparameter scaling. Empirically, our results suggest that SGLD–Gibbs achieves better-calibrated posterior uncertainty and better predictive

performance than variational methods.

There are a number of limitations of our work that motivate directions for future research. Our results are for latent variable models in which each data point is associated with a single local latent variable refreshed through Gibbs updates. Many important models exhibit more complex dependency structures, such as Bayesian matrix factorization, mixed-effects models, or hierarchical topic models, where latent variables are shared across observations or interact globally. Extending the joint scaling-limit framework to such settings would require capturing richer coupling between latent variables and global parameters, and may lead to qualitatively different limiting dynamics and hence distinct tuning guidance. Moreover, when the latent-variable updates do not mix rapidly, for instance under SGLD or Gibbs samplers with more complicated dependency structures, the resulting latent variables can exhibit temporal dependence and nontrivial interactions with the global-parameter iterates. Analyzing such regimes is an interesting direction for future work. Beyond these model-specific extensions, another important direction is to compare scaling-limit-based tuning with more algorithmic tuning criteria, such as the KSD-based strategy of Coullon et al. (2023).

## Impact Statement

This paper presents work whose goal is to advance the field of probabilistic machine learning. There are many potential societal consequences of our work, none of which we feel must be specifically highlighted here.

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

# A. Preliminaries for Proof of Main Result

## A.1. Assumptions

Recall that we assume throughout that $X_i \sim P$ independently for all $i \in \mathbb{N}$. We denote $\ell(\theta; X, z) := \log p(X, z \mid \theta)$.

**Assumption A.1.** $\nabla \log \pi_0$ is $L_0$-Lipschitz, and $\log p(x, z \mid \cdot) \in C^2(\Theta)$ for each $x, z \in (\mathcal{X}, \mathbb{R}^m)$

**Assumption A.2.** The exponents satisfy $\mathfrak{h} - \mathfrak{w} - \mathfrak{a}/3 > 0$ and $\mathbb{E}[\|\nabla \log p(X_1, \cdot \mid \theta^*)\|_\infty^{p_2}] < \infty$ for some $p_2 > \frac{1}{\mathfrak{h} - \mathfrak{w} - \mathfrak{a}/3}$.

**Assumption A.3.** For some $q_3 \in [0, \mathfrak{w}]$ and $p_3 := \frac{1}{\mathfrak{h} + q_3 - \mathfrak{w} - \mathfrak{a}/3}$, the local critical points satisfy $\|\hat{\theta}^{(n)} - \theta^*\| \in o_p(1/n^{q_3})$, and $\mathbb{E}[\|\nabla^{\otimes 2} \log(X_1, \cdot \mid \cdot)\|^{p_3}] < \infty$.

Let

$$\ell(\theta; x_i) := \log p(x_i \mid \theta) = \log \int p(x_i, z_i \mid \theta) \, dz_i$$

denote the log-likelihood function with the latent variable $z_i$ marginalized out. For any $r > 0$, let

$$B^{(n)}(r) := B(\hat{\theta}_n, r/n^{\mathfrak{w}})$$

denote the ball centered at the MLE $\hat{\theta}_n$ with radius $r/n^{\mathfrak{w}}$, for some scaling exponent $\alpha > 0$.

**Assumption A.4.** There is a non-decreasing sequence $r_{J,n} \to \infty$ such that

$$\sup_{\theta \in B(r_{J,n})} \|\frac{1}{n} \sum_{i=1}^{n_m} \nabla_\theta^{\otimes 2} \ell(\theta; X_i) + J_*\| \to 0$$

**Assumption A.5.** There is a non-decreasing sequence $r_{I,n} \to \infty$ such that

$$\sup_{\theta \in B(r_{I,n})} \|\frac{1}{n} \sum_{i=1}^{n_m} \mathbb{E}_{z|\theta,X}[\nabla_\theta \log p(X_i, z_i \mid \theta)^{\otimes 2}] - \tilde{I}_\star\| \to 0$$

**Assumption A.6.** If $\theta_n \to \theta^*$ when $n \to \infty$, then for almost every $X$ and $z$,

$$p(z|X, \theta_n) \to p(z|X, \theta^*)$$

when $n \to \infty$.

## A.2. Technical Lemmas

We will make use of the following two technical results in our proof.

**Proposition A.7** (Approximation of Markov chains (Ethier & Kurtz, 2009))**.** *Let*

$$A : C_c^\infty(\mathbb{R}^d) \to C(\mathbb{R}^d)$$

*be a linear operator, and suppose that the closure of the graph of $A$ with respect to the graph norm*

$$\|f\|_A := \|f\|_\infty + \|Af\|_\infty, \qquad f \in C_c^\infty(\mathbb{R}^d),$$

*generates a Feller semigroup $(T_t)_{t \geq 0}$ on $\mathbb{R}^d$. Let $(\theta_t)_{t \geq 0}$ be a Markov process with forward operator semigroup $(T_t)_{t \geq 0}$. Let $\{(\theta_k^{(n)})_{k \in \mathbb{N} \cup \{0\}}\}_{n \in \mathbb{N}}$ be a sequence of discrete-time Markov chains on $\mathbb{R}^d$ with respective transition kernels $\{U^{(n)}\}_{n \in \mathbb{N}}$. Suppose that $0 < \alpha^{(n)} \to \infty$, and define*

$$A^{(n)} := \alpha^{(n)}(U^{(n)} - I), \qquad T_t^{(n)} := (U^{(n)})^{\lfloor \alpha^{(n)} t \rfloor}, \qquad \theta_t^{(n)} := \theta_{\lfloor \alpha^{(n)} t \rfloor}^{(n)}.$$

*If*

$$\|A^{(n)} f - Af\|_\infty \longrightarrow 0 \quad \text{for all } f \in C_c^\infty(\mathbb{R}^d),$$

*then*

(a) $T_t^{(n)} \to T_t$ for each $t > 0$ and

(b) if $\theta^{(n)}(0) \Rightarrow \theta(0)$, then

$$\theta^{(n)}(\cdot) \Rightarrow \theta(\cdot) \quad \text{in the Skorokhod topology.}$$

**Lemma A.8** (Negrea et al. (2023)). *Let $(\Omega, \mathcal{F}, \mathbb{P})$ be a probability space, let $(\mathcal{X}, \tau)$ be a topological space endowed with the $\sigma$-field $\mathcal{F}_\mathcal{X} := \sigma(\tau)$, let $(X_n)_{n \in \mathbb{N}}$ be a sequence of $\mathcal{X}$-valued random elements, and let $x \in \mathcal{X}$. If for every subsequence $(n_m)$ there exists a sub-subsequence $(n_{m_k})$ such that*

$$X_{n_{m_k}} \longrightarrow x \quad \text{almost surely as } k \to \infty,$$

*then*

$$X_n \xrightarrow{\mathbb{P}} x.$$

*If $(\mathcal{X}, \tau)$ is first countable, then the converse also holds: if*

$$X_n \xrightarrow{\mathbb{P}} x,$$

*then for every subsequence $(n_m)$ there exists a sub-subsequence $(n_{m_k})$ such that*

$$X_{n_{m_k}} \longrightarrow x \quad \text{almost surely as } k \to \infty.$$

### A.3. Reduction to almost-sure convergence along subsequences

Define the random quantities

$$\Phi^{(n)} := \max\left\{\Phi_1^{(n)}, \Phi_2^{(n)}, \Phi_3^{(n)}\right\},$$

$$\Phi_1^{(n)} := n^{q_3} \|\hat{\theta}^{(n)} - \theta^\star\|,$$

$$\Phi_2^{(n)} := \sup_{\theta \in B(r_{J,n})} \left\| \frac{1}{n} \sum_{i=1}^{n} \nabla_\theta^{\otimes 2} \ell(\theta; X_i) + J_\star \right\|,$$

$$\Phi_3^{(n)} := \sup_{\theta \in B(r_{I,n})} \left\| \frac{1}{n} \sum_{i=1}^{n} \mathbb{E}_{Z_i | X_i, \theta}\left[ \nabla_\theta \log p(X_i, Z_i \mid \theta)^{\otimes 2} \right] - \tilde{I}_\star \right\|.$$

By assumption, $\Phi^{(n)} \xrightarrow{\mathbb{P}} 0$. By Lemma 1 of Negrea et al. (2023), for every subsequence $(n_m)_{m \in \mathbb{N}}$ there exists a further subsequence $(n_{m_k})_{k \in \mathbb{N}}$ such that

$$\Phi^{(n_{m_k})} \xrightarrow{\text{a.s.}} 0.$$

It therefore suffices to establish weak convergence of $\vartheta^{(n)}$ along any subsequence for which $\Phi^{(n)} \to 0$ almost surely.

Fix such a subsequence $(n_m)$ and define the event

$$\Omega^{(0)} := \bigcap_{j=1}^{3} \Omega^{(j)},$$

where

$$\Omega^{(1)} := \left\{ \Phi^{(n_m)} \to 0 \right\},$$

$$\Omega^{(2)} := \left\{ \max_{1 \leq i \leq n} \|\nabla \ell(\theta^*; X_i, \cdot)\| \leq n^{1/p_2} \text{ a.b.f.o.} \right\},$$

$$\Omega^{(3)} := \left\{ \max_{1 \leq i \leq n} \|\nabla^{\otimes 2} \ell(\cdot; X_i, \cdot)\|_\infty \leq n^{1/p_3} \text{ a.b.f.o.} \right\}.$$

By the assumed moment conditions and Lemma 2 of Negrea et al. (2023) applied to the power functions $t \mapsto t^{p_2}$ and $t \mapsto t^{p_3}$, the event $\Omega^{(0)}$ has probability one. We will show that on $\Omega^{(0)}$, all remainder terms appearing in the generator expansions are negligible, and the convergence of the discrete-time Markov generators to their continuous-time limit follows.

# B. Proof of Main Theorem

## B.1. Overview

Our proof follows the spirit of Negrea et al. (2023). We establish weak convergence of the processes in the Skorokhod topology in probability by first proving almost sure convergence along subsequences. This is achieved by showing that the difference between the approximate generator and the limiting generator, evaluated on smooth test functions with compact support, vanishes uniformly. Using Lemma A.8, this then yields weak convergence in the Skorokhod topology in probability. As in Negrea et al. (2023), we divide the proof into two parts for technical reasons.

In Part 1, we consider arguments that are sufficiently far from the support of the test function. The main idea is to control the probability that the global-parameter process jumps back into the support of the test function. A key difference with Negrea et al. (2023) is that we must impose assumptions on the joint likelihood uniformly on latent variable values to ensure that, even in the presence of additional uncertainty induced by the latent-variable updates, the probability that the global parameter jumps back into the support can still be controlled at a comparable scale.

In Part 2, we analyze arguments that lie in or are close to the support of the test function. We perform a Taylor expansion of the joint approximate generator. In Section B.5, as a result of Equation (B.1), the drift term converges in the same way as in Negrea et al. (2023). In Section B.6, for the gradient component of the diffusion term, a similar approach applies, but the resulting limit now explicitly incorporates additional variability arising from sampling the latent variables. In Section B.7, we introduce a new term that bridges the infinitesimal operator associated with Gibbs updates and the generator of a Poisson jump process. The most technically challenging new aspect is in Sections B.5 and B.9, showing that all cross terms involving both the global parameters and the latent variables vanish in the limit. This vanishing further implies that, in the asymptotic regime, the contribution of any single observation becomes asymptotically independent of the global-parameter dynamics, which in turn allows the joint limiting distribution to factorize into independent components.

## B.2. Notation useful for the proof

We first introduce notation for the increments of the localized algorithm. Throughout, we condition on $\vartheta_0^{(n)} = \vartheta$ and $\zeta_0^{(n)} = \zeta$, and write $\zeta_1^{(n)} = \tilde{\zeta}$ for the updated latent variable.

Define the following components of the one-step increment:

$$\Delta_\xi^{(n)} := w^{(n)}\sqrt{h^{(n)}(\beta^{(n)})^{-1}\Gamma}\,\xi_1,$$

$$\Delta_{\pi_0}^{(n)} := \frac{h^{(n)}w^{(n)}\Gamma}{2n}\nabla \log \pi_0\Big(\hat{\theta}^{(n)} + (w^{(n)})^{-1}\vartheta\Big),$$

$$\Delta_\ell^{(n)} := \frac{h^{(n)}w^{(n)}\Gamma}{2b^{(n)}}\sum_{j=1}^{b^{(n)}}\nabla_\theta \ell\Big(\hat{\theta}^{(n)} + (w^{(n)})^{-1}\vartheta; X_{I_1^{(n)}(j)}, \tilde{\zeta}_{I_1^{(n)}(j)}\Big),$$

and set

$$\Delta^{(n)} = \Delta_\xi^{(n)} + \Delta_{\pi_0}^{(n)} + \Delta_\ell^{(n)}.$$

The latent variables are updated according to

$$\tilde{\zeta}_i \sim p\Big(z \mid X_i, \hat{\theta}^{(n)} + (w^{(n)})^{-1}\vartheta\Big), \qquad i = 1, \dots, n,$$

independently conditional on the observations and the current parameter value.

We next define a sequence of generator-like operators acting on test functions $f$ by

$$A^{(n)}f(\vartheta, \zeta_1) := \alpha^{(n)}\Big(\mathbb{E}\big[f(\vartheta + \Delta^{(n)}, \tilde{\zeta}_1)\big] - f(\vartheta, \zeta_1)\Big),$$

where the expectation is taken over all algorithmic randomness, including minibatch sampling, Gaussian noise, and Gibbs updates, conditional on the observations.

For sufficiently smooth test functions $f$, the generator $A$ of the limiting Lévy process is given by

$$(Af)(\vartheta, \zeta_1) = -\langle B\vartheta, \nabla_\vartheta f(\vartheta, \zeta_1) \rangle + \frac{1}{2} A : \nabla_\vartheta^{\otimes 2} f(\vartheta, \zeta_1)$$
$$+ \lambda \left( \int f(\vartheta, z) \, p(z \mid X_1, \theta^\star) \, dz - f(\vartheta, \zeta_1) \right),$$

with

$$B = c_h \Gamma J_\star \mathbf{1}\{\mathfrak{a} = \mathfrak{h}\},$$
$$A = \frac{c_h}{c_\beta} \Gamma \mathbf{1}\{\mathfrak{h} + \mathfrak{b} \le \mathfrak{t}\} + \frac{c_h^2}{4 c_b} \Gamma \tilde{I}_\star \Gamma^\top \mathbf{1}\{\mathfrak{t} \le \mathfrak{b} + \mathfrak{h}\}$$
$$\lambda = c_b.$$

Consider realization of $X^{(n)} \in \Omega^{(n)}$, we want to show that for all $f \in C_c^\infty(\mathbb{R}^{d+s})$ and any $\zeta_1$,

$$\lim_{m \to \infty} \sup_{\vartheta \in \mathbb{R}^d} \|A^{(n_m)} f(\vartheta, \zeta_1) - Af(\vartheta, \zeta_1)\| = 0.$$

We will show this in two parts. To begin, note that, for any test function $f$ with compact support, there exists $K_0$ such that $f(\theta) = 0$ for all $\theta \in K_0^c$. First we will identify a extension set $K_1 \subset K_0$ such that

$$\lim_{m \to \infty} \sup_{\vartheta \in K_1^c} \|A^{(n_m)} f(\vartheta, \zeta_1) - Af(\vartheta, \zeta_1)\| = 0.$$

Second, we will show that

$$\lim_{m \to \infty} \sup_{\vartheta \in K_1} \|A^{(n_m)} f(\vartheta, \zeta_1) - Af(\vartheta, \zeta_1)\| = 0.$$

### B.3. Part 1.

For all $\vartheta \in K_0^c$, we have $f(\vartheta) = 0$, $\nabla_\vartheta f(\vartheta) = 0$, and $\nabla_\vartheta^{\otimes 2} f(\vartheta) = 0$. Therefore, for any $K_1 \supseteq K_0$,

$$\sup_{\vartheta \in K_1^c} \|A^{(n_m)} f(\vartheta, \zeta_1) - Af(\vartheta, \zeta_1)\| = \alpha^{(n_m)} \sup_{\vartheta \in K_1^c} \mathbb{E}[f(\vartheta + \Delta^{(n_m)}(\vartheta), \tilde{\zeta}_1)]$$
$$\le \alpha^{(n_m)} \|f\|_\infty \sup_{\vartheta \in K_1^c} \mathbb{P}[\vartheta + \Delta^{(n_m)}(\vartheta) \in K_0].$$

Let

$$K_1 := \{\vartheta : \|\vartheta\| \le 2R_0 + 2c_0\}, \qquad R_0 := \sup_{\vartheta \in K_0} \|\vartheta\|,$$

where

$$c_0 := \frac{c_h}{2} (3 + \|\Gamma \nabla \log \pi_0(\theta^*)\|) + \sqrt{c_h / c_\beta \Gamma}.$$

Then, for $\vartheta \in K_1^c$, under the assumption that $\Gamma \nabla \log \pi_0$ is $L_0$-Lipschitz,

$$\|\Delta_{\pi_0}^{(n_m)}(\vartheta)\| = \frac{h^{(n_m)} w^{(n_m)} \Gamma}{2n} \|\nabla \log \pi_0(\hat{\theta}^{(n_m)} + (w^{(n_m)})^{-1} \vartheta)\|$$
$$\le \frac{c_h n^{\mathfrak{w} - \mathfrak{h} - 1}}{2} \left( \|\Gamma \nabla \log \pi_0(\theta^*)\| + L_0 \|\hat{\theta}^{(n_m)} - \theta^*\| + \frac{L_0 \|\vartheta\|}{n^{\mathfrak{w}}} \right).$$

Similarly,

$$
\begin{aligned}
\|\Delta_\ell^{(n_m)}(\vartheta)\| &= \frac{h^{(n_m)} w^{(n_m)} \|\Gamma\|}{2 b^{(n_m)}} \Big\| \sum_{i=1}^{b^{(n_m)}} \nabla \ell(\hat{\theta}^{(n_m)} + (w^{(n_m)})^{-1} \vartheta; X_{I_1^{(n)}(i)}, \tilde{\zeta}_{I_1^{(n)}(i)}) \Big\| \\
&\leq \frac{c_h n^{\mathfrak{w}-\mathfrak{h}} \|\Gamma\|}{2 b^{(n_m)}} \sum_{i \in [n_m]} \|\nabla \ell(\theta^*; X_{I_1^{(n_m)}(i)}, \tilde{\zeta}_{I_1^{(n_m)}(i)})\| \\
&\quad + L(X_{I_1^{(n_m)}(i)}, \tilde{\zeta}_{I_1^{(n_m)}(i)}) \|\hat{\theta}^{(n_m)} - \theta^*\| \\
&\quad + L(X_{I_1^{(n_m)}(i)}, \tilde{\zeta}_{I_1^{(n_m)}(i)}) \frac{\|\vartheta\|}{n_m^{\mathfrak{w}}}.
\end{aligned}
$$

We define the (random) Lipschitz constants

$$
\begin{aligned}
L(X_i) &:= \|\nabla^{\otimes 2} \ell(\cdot; X_i, \cdot)\|_\infty, \\
L_*(X^{(n_m)}) &:= \max_{i \in [n_m]} \|\nabla \ell(\theta^*; X_i, \cdot)\|, \\
L(X^{(n_m)}, \tilde{\zeta}^{(n_m)}) &:= \max_{i \in [n_m]} L(X_i).
\end{aligned}
$$

Since $X^{(n_m)} \in \Omega^{(0)}$, we have that $\Phi^{(n_m)} \to 0$, $\max_{i \in [n_m]} \|\nabla \ell(\theta^*; X_i, \cdot)\|_\infty \leq n_m^{1/p_2}$, and $\max_{i \in [n_m]} \|\nabla^{\otimes 2} \ell(\cdot; X_i, \cdot)\|_\infty \leq n_m^{1/p_3}$. Thus if $m$ is large enough that all of the following hold:

$$
\sup_{m' \geq m} \Phi^{(n_m)} \leq \min(1, L_0^{-1}),
$$

$$
1 \geq \sup_{m' \geq m} \frac{L_*(X^{(n_{m'})})}{n_{m'}^{1/p_2}}
$$

$$
n_m \geq \max\left( (2 c_h \|\Gamma\|)^{1/(1/p_3 - \mathfrak{h})}, (2 c_h L_0 \|\Gamma\|)^{\frac{1}{\mathfrak{h}+1-\mathfrak{a}-\mathfrak{w}}} \right)
$$

$$
1 \geq \sup_{m' \geq m} \frac{L(X^{(n_{m'})})}{n_{m'}^{1/p_3}}.
$$

Then, using that $0 < \mathfrak{w} < 1$,

$$
\|\Delta_{\pi_0}^{(n_m)}(\vartheta)\| \leq \frac{c_h \|\Gamma\|}{2} \big( \|\nabla \log \pi_0(\theta^*)\| + 1 \big) + \frac{1}{4} \|\vartheta\|, \qquad \|\Delta_\ell^{(n_m)}(\vartheta)\| \leq c_h \|\Gamma\| + \frac{1}{4} \|\vartheta\|.
$$

Therefore, for $\vartheta \in K_1^c$,

$$
\|\vartheta\| - \|\Delta_{\pi_0}^{(n_m)}(\vartheta)\| - \|\Delta_\ell^{(n_m)}(\vartheta)\| - R_0 \geq \frac{1}{2} \|\vartheta\| - \frac{c_h \|\Gamma\|}{2} (3 + \|\nabla \log \pi_0(\theta^*)\|) - R_0 \geq \sqrt{c_h \|\Gamma\|/c_\beta}.
$$

Consequently,

$$
\lim_{m \to \infty} \sup_{\vartheta \in K_1^c} \|A^{(n_m)} f(\vartheta, \zeta_1) - A f(\vartheta, \zeta_1)\| \leq \lim_{m \to \infty} \alpha^{n_m} \|f\|_\infty \mathbb{P}\Big( \|\xi_1\| \geq n_m^{\mathfrak{h}/2 + \mathfrak{t}/2 - \mathfrak{w}} \Big) = 0.
$$

## B.4. Part 2.

We take a partial second-order Taylor expansion of the test function $f$ with respect to the global variable $\vartheta$:

$$
\begin{aligned}
A^{(n_m)}f(\vartheta,\zeta_1) &= \alpha^{(n_m)}\big(\mathbb{E}[f(\vartheta+\Delta^{(n_m)}(\vartheta),\tilde{\zeta}_1)] - f(\vartheta,\zeta_1)\big) \\
&= \alpha^{(n_m)}\Big(\mathbb{E}[f(\vartheta+\Delta^{(n_m)}(\vartheta),\tilde{\zeta}_1) - f(\vartheta,\tilde{\zeta}_1)] + \mathbb{E}[f(\vartheta,\tilde{\zeta}_1) - f(\vartheta,\zeta_1)]\Big) \\
&= n_m^{\mathfrak{a}}\mathbb{E}\Big\langle \nabla_\vartheta f(\vartheta,\tilde{\zeta}_1),\Delta^{(n_m)}(\vartheta)\Big\rangle + n_m^{\mathfrak{a}}\mathbb{E}\Big\langle \tfrac{1}{2}\nabla_\vartheta^{\otimes 2}f(\vartheta,\tilde{\zeta}_1)\Delta^{(n_m)}(\vartheta),\Delta^{(n_m)}(\vartheta)\Big\rangle \\
&\quad + n_m^{\mathfrak{a}}\mathbb{E}\Big[\tfrac{1}{6}\nabla_\vartheta^{\otimes 3}f(\vartheta+S\Delta^{(n_m)}(\vartheta),\tilde{\zeta}_1)(\Delta^{(n_m)}(\vartheta),\Delta^{(n_m)}(\vartheta),\Delta^{(n_m)}(\vartheta))\Big] \\
&\quad + \mathbb{E}[f(\vartheta,\tilde{\zeta}_1) - f(\vartheta,\zeta_1)].
\end{aligned}
$$

Rearranging terms yields

$$
\begin{aligned}
A^{(n_m)}f(\vartheta,\zeta_1) &= n_m^{\mathfrak{a}}\mathbb{E}\Big\langle \nabla_\vartheta f(\vartheta,\zeta_1),\Delta^{(n_m)}(\vartheta)\Big\rangle + n_m^{\mathfrak{a}}\mathbb{E}\Big\langle \tfrac{1}{2}\nabla_\vartheta^{\otimes 2}f(\vartheta,\zeta_1)\Delta^{(n_m)}(\vartheta),\Delta^{(n_m)}(\vartheta)\Big\rangle \\
&\quad + n_m^{\mathfrak{a}}\mathbb{E}\Big[\tfrac{1}{6}\nabla_\vartheta^{\otimes 3}f(\vartheta+S\Delta^{(n_m)}(\vartheta),\tilde{\zeta}_1)(\Delta^{(n_m)}(\vartheta),\Delta^{(n_m)}(\vartheta),\Delta^{(n_m)}(\vartheta))\Big] \\
&\quad + \mathbb{E}[f(\vartheta,\tilde{\zeta}_1) - f(\vartheta,\zeta_1)] \\
&\quad + n_m^{\mathfrak{a}}\mathbb{E}\Big\langle \nabla_\vartheta f(\vartheta,\tilde{\zeta}_1) - \nabla_\vartheta f(\vartheta,\zeta_1),\Delta^{(n_m)}(\vartheta)\Big\rangle \\
&\quad + n_m^{\mathfrak{a}}\mathbb{E}\Big\langle \tfrac{1}{2}\nabla_\vartheta^{\otimes 2}f(\vartheta,\tilde{\zeta}_1)\Delta^{(n_m)}(\vartheta) - \tfrac{1}{2}\nabla_\vartheta^{\otimes 2}f(\vartheta,\zeta_1)\Delta^{(n_m)}(\vartheta),\Delta^{(n_m)}(\vartheta)\Big\rangle
\end{aligned}
$$

for some $S \in [0,1]$. Therefore,

$$
\begin{aligned}
\|A^{(n_m)}f - Af\| &\leq \underbrace{\left\| n_m^{\mathfrak{a}}\mathbb{E}\Big\langle \nabla_\vartheta f(\vartheta,\zeta_1),\Delta^{(n_m)}(\vartheta)\Big\rangle + \Big\langle \tfrac{1}{2}B\vartheta,\nabla_\vartheta f(\vartheta,\zeta_1)\Big\rangle \right\|}_{R_1} \\
&\quad + \underbrace{\left\| n_m^{\mathfrak{a}}\mathbb{E}\Big\langle \tfrac{1}{2}\nabla_\vartheta^{\otimes 2}f(\vartheta,\zeta_1)\Delta^{(n_m)}(\vartheta),\Delta^{(n_m)}(\vartheta)\Big\rangle - \tfrac{1}{2}A:\nabla_\vartheta^{\otimes 2}f(\vartheta,\zeta_1) \right\|}_{R_2} \\
&\quad + \underbrace{\left\| \lambda\Big(\int f(\vartheta,y)p(y|X_1,\theta^*)\,dy - f(\vartheta,\zeta_1)\Big) - n_m^{\mathfrak{a}}\mathbb{E}[f(\vartheta,\tilde{\zeta}_1) - f(\vartheta,\zeta_1)] \right\|}_{R_3} \\
&\quad + \underbrace{\left\| n_m^{\mathfrak{a}}\mathbb{E}\Big\langle \nabla_\vartheta f(\vartheta,\tilde{\zeta}_1) - \nabla_\vartheta f(\vartheta,\zeta_1),\Delta^{(n_m)}(\vartheta)\Big\rangle \right\|}_{R_4} \\
&\quad + \underbrace{\left\| n_m^{\mathfrak{a}}\mathbb{E}\Big\langle \tfrac{1}{2}\nabla_\vartheta^{\otimes 2}f(\vartheta,\tilde{\zeta}_1)\Delta^{(n_m)}(\vartheta) - \tfrac{1}{2}\nabla_\vartheta^{\otimes 2}f(\vartheta,\zeta_1)\Delta^{(n_m)}(\vartheta),\Delta^{(n_m)}(\vartheta)\Big\rangle \right\|}_{R_5} \\
&\quad + \underbrace{\left\| n_m^{\mathfrak{a}}\mathbb{E}\Big[\tfrac{1}{6}\nabla_\vartheta^{\otimes 3}f(\vartheta+S\Delta^{(n_m)}(\vartheta),\tilde{\zeta}_1)(\Delta^{(n_m)}(\vartheta),\Delta^{(n_m)}(\vartheta),\Delta^{(n_m)}(\vartheta))\Big] \right\|}_{R_6}.
\end{aligned}
$$

We denote the six terms above by $R_1,\ldots,R_6$ and will show that each of them vanishes as $n \to \infty$.

## B.5. $R_1$ (drift term)

We have

$$
n_m^{\mathfrak{a}}\,\mathbb{E}[\Delta^{(n_m)}(\vartheta)] = n_m^{\mathfrak{a}}\,\mathbb{E}\big[\Delta_\xi^{(n_m)}(\vartheta) + \Delta_{\pi_0}^{(n_m)}(\vartheta) + \Delta_\ell^{(n_m)}(\vartheta)\big].
$$

**Noise term.**

$$n_m^{\mathfrak{a}}\,\mathbb{E}[\Delta_\xi^{(n_m)}(\vartheta)] = n_m^{\mathfrak{a}}\,\mathbb{E}\left[w^{(n_m)}\sqrt{h^{(n_m)}(\beta)^{-1}\Gamma}\,\xi_1\right] = n_m^{\mathfrak{a}}\,w^{(n_m)}\sqrt{h^{(n_m)}(\beta)^{-1}\Gamma}\,\mathbb{E}[\xi_1] = 0.$$

**Prior term.**

$$\begin{aligned}
n_m^{\mathfrak{a}}\,\mathbb{E}[\Delta_{\pi_0}^{(n_m)}(\vartheta)] &= n_m^{\mathfrak{a}}\,\mathbb{E}\left[\frac{h^{(n_m)}w^{(n_m)}\Gamma}{2n}\,\nabla\log\pi_0\big(\hat{\theta}^{(n_m)} + (w^{(n_m)})^{-1}\vartheta\big)\right]\\
&= \frac{c_h}{2}\,n_m^{\mathfrak{a}-\mathfrak{h}+\mathfrak{w}-1}\,\Gamma\,\nabla\log\pi_0\big(\hat{\theta}^{(n_m)} + (w^{(n_m)})^{-1}\vartheta\big)\\
&= \frac{c_h n_m^{\mathfrak{a}-\mathfrak{h}+\mathfrak{w}-1}\|\Gamma\|}{2}\left(\nabla\log\pi_0(\theta^*) + L_0(\Phi^{(n_m)} + \frac{2R_0 + 2c_0}{n_m^{\mathfrak{w}}})\right).
\end{aligned}$$

Which vanishes uniformly on $K_1$, as $\mathfrak{a} - \mathfrak{h} + \mathfrak{w} - 1 < 0$

**Likelihood term.**

$$\begin{aligned}
n_m^{\mathfrak{a}}\mathbb{E}\Delta_\ell^{(n_m)}(\vartheta) &= n^{\mathfrak{a}}\mathbb{E}\frac{h^{(n_m)}w^{(n_m)}\Gamma}{2b^{(n_m)}}\sum_{i=1}^{b^{(n_m)}}\nabla\ell(\hat{\theta}^{(n_m)} + (w^{(n_m)})^{-1}\vartheta; X_{I_1^{(n)}(i)}, \tilde{\zeta}_{I_1^{(n)}(i)})\\
&= \frac{c_h n^{\mathfrak{a}-\mathfrak{h}+\mathfrak{w}-1}\Gamma}{2}\mathbb{E}_{\tilde{\zeta}}\sum_{i=1}^{n_m}\nabla\ell(\hat{\theta}^{(n_m)} + n_m^{-\mathfrak{w}}\vartheta; X_i, \tilde{\zeta}_i)\\
&= \frac{c_h n_m^{\mathfrak{a}-\mathfrak{h}+\mathfrak{w}-1}\Gamma}{2}\sum_{i=1}^{n_m}\nabla\ell(\hat{\theta}^{(n_m)} + n^{-\mathfrak{w}}\vartheta; X_i)\\
&= \frac{c_h n_m^{\mathfrak{a}-\mathfrak{h}+\mathfrak{w}-1}\Gamma}{2}\sum_{i=1}^{n_m}[\nabla\ell(\hat{\theta}^{(n_m)}; X_i) + n^{-\mathfrak{w}}\nabla^2\ell(\hat{\theta}^{(n_m)}; X_i)\vartheta]\\
&= \frac{c_h n_m^{\mathfrak{a}-\mathfrak{h}}\Gamma}{2}\frac{\vartheta}{n_m}\sum_{i=1}^{n_m}\int_0^1\nabla^{\otimes 2}\ell(\hat{\theta}^{(n_m)} + \frac{s}{n_m^{\mathfrak{w}}\vartheta}; X_i)ds
\end{aligned}$$

where we used that $\ell(\theta; X_i)$ denotes the (marginal) log-likelihood with $z$ marginalized, and

$$\mathbb{E}_{\tilde{\zeta}}[g(\tilde{\zeta}_i)] = \int g(\tilde{\zeta}_i)\,p(\tilde{\zeta}_i \mid X_i, \hat{\theta}^{(n_m)} + n_m^{-\mathfrak{w}}\vartheta)\,d\tilde{\zeta}_i.$$

Indeed,

$$\begin{aligned}
\mathbb{E}_{\tilde{\zeta}}\left[\nabla\ell(\hat{\theta}^{(n_m)} + n_m^{-\mathfrak{w}}\vartheta; X_i, \tilde{\zeta}_i)\right] &= \int\nabla\log p(X_i, \tilde{\zeta}_i \mid \hat{\theta}^{(n_m)} + n_m^{-\mathfrak{w}}\vartheta)\,p(\tilde{\zeta}_i \mid X_i, \hat{\theta}^{(n_m)} + n_m^{-\mathfrak{w}}\vartheta)\,d\tilde{\zeta}_i &\text{(B.1)}\\
&= \frac{\nabla p(X_i \mid \hat{\theta}^{(n_m)} + n_m^{-\mathfrak{w}}\vartheta)}{p(X_i \mid \hat{\theta}^{(n_m)} + n_m^{-\mathfrak{w}}\vartheta)}\\
&= \nabla\ell(\hat{\theta}^{(n_m)} + n_m^{-\mathfrak{w}}\vartheta; X_i).
\end{aligned}$$

Moreover, by the definition of $\hat{\theta}^{(n_m)}$,

$$\sum_{i=1}^{n_m}\nabla\ell(\hat{\theta}^{(n_m)}; X_i) = 0.$$

Hence, when $\mathfrak{a} + \mathfrak{h} < 1$, $n_m^{\mathfrak{a}}\mathbb{E}\Delta_\ell^{(n_m)}(\vartheta)$ vanishes and the drift term will be inactive in the limit.

When $\mathfrak{a} = \mathfrak{h}$, and $n_m$ large enough that $r_{J,n_m} \geq R_0 + c_0$,

$$R_1 = \left\|n_m^{\mathfrak{a}}\mathbb{E}\left\langle\nabla_\vartheta f(\vartheta, \zeta_1), \Delta^{(n_m)}(\vartheta)\right\rangle + \left\langle\frac{c_h\Gamma J_\star}{2}\vartheta, \nabla_\vartheta f(\vartheta, \zeta_1)\right\rangle\right\| \leq c_h\|f\|_\infty\|\Gamma\|(R_0 + c_0)\Phi^{(n_m)},$$

which vanishes uniformly on $K_1$

**B.6.** $R_2$ **(diffusion term)**

We write

$$n_m^{\mathfrak{a}} \, \mathbb{E}\Big[(\Delta^{(n_m)}(\vartheta))^{\otimes 2}\Big] = n_m^{\mathfrak{a}} \, \mathbb{E}\Big[(\Delta_\xi^{(n_m)}(\vartheta) + \Delta_{\pi_0}^{(n_m)}(\vartheta) + \Delta_\ell^{(n_m)}(\vartheta))^{\otimes 2}\Big].$$

Given the independence among all three terms, the cross term will vanish. So the potentially non-vanishing contributions come from $n_m^{\mathfrak{a}} \mathbb{E}\Big[(\Delta_\xi^{(n_m)}(\vartheta))^{\otimes 2}\Big]$ and $n_m^{\mathfrak{a}} \mathbb{E}\Big[(\Delta_\ell^{(n_m)}(\vartheta))^{\otimes 2}\Big]$.

To obtain a non-trivial limit, we only considering $\mathfrak{a} = \mathfrak{h}$.

**Noise variance.**

$$n_m^{\mathfrak{a}} \, \mathbb{E}\Big[(\Delta_\xi^{(n_m)}(\vartheta))^{\otimes 2}\Big] = \frac{c_h}{c_\beta} \, n_m^{\mathfrak{a}+2\mathfrak{w}-\mathfrak{h}-\mathfrak{t}} \, \Gamma = \frac{c_h}{c_\beta} \, n_m^{2\mathfrak{w}-\mathfrak{h}} \, \Gamma.$$

Thus when $\mathfrak{w} < \mathfrak{t}/2$, the corresponding diffusion term is inactive in the limit. When $\mathfrak{w} = \mathfrak{t}/2$,

$$\|n_m^{\mathfrak{a}} \, \mathbb{E}\Big[(\Delta_\xi^{(n_m)}(\vartheta))^{\otimes 2}\Big] - \frac{c_h \Gamma}{c_\beta} \| = 0.$$

**Stochastic gradient variance.**

$$
\begin{aligned}
n^{\mathfrak{a}} \mathbb{E}[\Delta_\ell^{(n_m)}(\vartheta)]^2 &= n^{\mathfrak{a}} \mathbb{E}\left[ \frac{h^{(n_m)} w^{(n_m)}}{2b^{(n_m)}} \sum_{i=1}^{b^{(n_m)}} \nabla\ell(\hat{\theta}^{(n_m)} + (w^{(n_m)})^{-1}\vartheta; X_{I_1^{(n)}(i)}, \tilde{\zeta}_{I_1^{(n)}(i)}) \right]^2 \\
&= \frac{c_h^2}{4c_b} n^{\mathfrak{a}-2\mathfrak{h}+2\mathfrak{w}-2\mathfrak{b}} \Gamma \mathbb{E}\left[ \sum_{i=1}^{b^{(n_m)}} \nabla\ell(\hat{\theta}^{(n_m)} + (w^{(n_m)})^{-1}\vartheta; X_{I_1^{(n)}(i)}, \tilde{\zeta}_{I_1^{(n)}(i)}) \right]^2 \Gamma' \\
&= \frac{c_h^2}{4c_b} n^{\mathfrak{a}-2\mathfrak{h}+2\mathfrak{w}-\mathfrak{b}} \Gamma \left[ \frac{1}{n_m} \mathbb{E}_{\tilde{\zeta}} \sum_{i=1}^{n_m} \nabla\ell(\hat{\theta}^{(n_m)} + (w^{(n_m)})^{-1}\vartheta; X_i, \tilde{\zeta}_i)^2 \right] \Gamma' \\
&\quad + \frac{c_h^2}{4c_b^2} n^{\mathfrak{a}-2\mathfrak{h}+2\mathfrak{w}-2\mathfrak{b}} \Gamma \left[ \mathbb{E} \sum_{i=1}^{b^{(n_m)}} \sum_{i'=1, i'\neq i}^{b^{(n_m)}} \nabla\ell(\hat{\theta}^{(n_m)} + (w^{(n_m)})^{-1}\vartheta; X_{I_1^{(n)}(i)}, \tilde{\zeta}_{I_1^{(n)}(i)}) \right. \\
&\qquad\qquad\qquad\qquad\qquad \left. \otimes \nabla\ell(\hat{\theta}^{(n_m)} + (w^{(n_m)})^{-1}\vartheta; X_{I_1^{(n)}(i')}, \tilde{\zeta}_{I_1^{(n)}(i')}) \right] \Gamma' \\
&= \frac{c_h^2}{4c_b} n_m^{\mathfrak{a}-2\mathfrak{h}+2\mathfrak{w}-\mathfrak{b}} \Gamma \left[ \frac{1}{n_m} \mathbb{E}_{\tilde{\zeta}} \sum_{i=1}^{n_m} \nabla\ell(\hat{\theta}^{(n_m)} + (w^{(n_m)})^{-1}\vartheta; X_i, \tilde{\zeta}_i)^2 \right] \Gamma' \\
&\quad + \frac{c_h^2}{4c_b^2} n_m^{\mathfrak{a}-2\mathfrak{h}+2\mathfrak{w}-2\mathfrak{b}} \frac{b^{(n_m)}(b^{(n_m)}-1)}{n_m^2} \Gamma \left[ \sum_{i=1}^{n_m} \sum_{i'=1}^{n_m} \nabla\ell(\hat{\theta}^{(n_m)} + (w^{(n_m)})^{-1}\vartheta; X_{I_1^{(n)}(i)}) \right. \\
&\qquad\qquad\qquad\qquad\qquad \left. \otimes \nabla\ell(\hat{\theta}^{(n_m)} + (w^{(n_m)})^{-1}\vartheta; X_{I_1^{(n)}(i')}) \right] \Gamma' \\
&= \frac{c_h^2}{4c_b} n_m^{\mathfrak{a}-2\mathfrak{h}+2\mathfrak{w}-\mathfrak{b}} \Gamma \left[ \frac{1}{n_m} \mathbb{E}_{\tilde{\zeta}} \sum_{i=1}^{n_m} \nabla\ell(\hat{\theta}^{(n_m)} + (w^{(n_m)})^{-1}\vartheta; X_i, \tilde{\zeta}_i)^2 \right] \Gamma' \\
&\quad + \frac{c_h^2}{4c_b^2} n_m^{\mathfrak{a}-2\mathfrak{h}+2\mathfrak{w}-2\mathfrak{b}} b^{(n_m)}(b^{(n_m)}-1) \Gamma \left( \frac{1}{n_m} \sum_{i=1}^{n_m} \int_0^1 \nabla^{\otimes 2}\ell(\hat{\theta}^{(n_m)} + \frac{s}{n_m^{\mathfrak{w}}}\vartheta; X_i) ds \frac{1}{n_m^{\mathfrak{w}}}\vartheta \right)^{\otimes 2} \Gamma'.
\end{aligned}
$$

Note that

$$\left\| \frac{c_h^2}{4c_b^2} n_m^{\mathfrak{a}-2\mathfrak{h}+2\mathfrak{w}-2\mathfrak{b}} b^{(n_m)}(b^{(n_m)}-1)\Gamma\left(\frac{1}{n_m}\sum_{i=1}^{n_m}\int_0^1 \nabla^{\otimes 2}\ell(\hat{\theta}^{(n_m)}+\frac{s}{n_m^{\mathfrak{w}}}\vartheta; X_i)ds\frac{1}{n_m^{\mathfrak{w}}}\vartheta\right)^{\otimes 2}\Gamma f'\right\|$$

$$\leq \sqrt{d}c_h^2 n_m^{\mathfrak{a}-2\mathfrak{h}+2\mathfrak{w}}\|\Gamma\|^2\|\nabla^{\otimes 2}f\|_\infty \frac{(2R_0+2c_0)^2}{n_m^{2\mathfrak{w}}}(J_\star+\Phi^{(n_m)})^2$$

Since $\mathfrak{a}=\mathfrak{h}$ and $\mathfrak{h}>0$, this term vanishes uniformly on $K_1$.

Thus when $\mathfrak{w}<(\mathfrak{h}+\mathfrak{b})/2$, the corresponding diffusion term is inactive in the limit. When $\mathfrak{w}<(\mathfrak{h}+\mathfrak{b})/2$ and $n_m$ large enough that $r_{I,n_m}\geq R_0+c_0$,

$$\|n^{\mathfrak{a}}\mathbb{E}[\Delta_\ell^{(n_m)}(\vartheta)]^2 - \frac{c_h^2}{4c_b}\Gamma\tilde{I}_\star\Gamma'\| \leq \frac{c_h^2}{4c_b}\|\Gamma\|^2\Phi^{(n_m)}$$

vanishes uniformly on $K_1$. Hence, $R_2$ vanishes uniformly on $K_1$.

## B.7. $R_3$ (jump term)

We consider

$$n_m^{\mathfrak{a}}\mathbb{E}\left[f(\vartheta,\tilde{\zeta}_1)-f(\vartheta,\zeta_1)\right] = c_b\, n_m^{\mathfrak{a}+\mathfrak{b}-1}\left[\int f(\vartheta,y)\,p(y\mid X_1,\hat{\theta}^{(n_m)}+n_m^{-\mathfrak{w}}\vartheta)\,d\zeta - f(\vartheta,\zeta_1)\right]$$

$$= c_b\, n_m^{\mathfrak{a}+\mathfrak{b}-1}\left[\int f(\vartheta,y)\,p(y\mid X_1,\theta^*)\,d\zeta - f(\vartheta,\zeta_1)\right]$$

$$+ c_b\, n_m^{\mathfrak{a}+\mathfrak{b}-1}\left[\int f(\vartheta,y)\left(p(y\mid X_1,\hat{\theta}^{(n_m)}+n_m^{-\mathfrak{w}}\vartheta)-p(y\mid X_1,\theta^*)\right)dy\right].$$

Since $\|\hat{\theta}^{(n_m)}+n_m^{-\mathfrak{w}}\vartheta-\theta^*\|\leq\|\hat{\theta}^{(n_m)}-\theta^*\|+n_m^{-\mathfrak{w}}\|\vartheta\|$ vanishes uniformly on $K_1$, under Theorem A.6, so do $\|p(y\mid X_1,\hat{\theta}^{(n_m)}+n_m^{-\mathfrak{w}}\vartheta)-p(y\mid X_1,\theta^*)\|$.

Thus when $\mathfrak{b}+\mathfrak{h}<1$, this term is inactive in the limit. When $\mathfrak{b}+\mathfrak{h}=1$,

$$R_3 = \|\lambda\left(\int f(\vartheta,y)p(y|X_1,\theta^*)\,dy - f(\vartheta,\zeta_1)\right) - n_m^{\mathfrak{a}}\mathbb{E}[f(\vartheta,\tilde{\zeta}_1)-f(\vartheta,\zeta_1)]\|$$

vanishes uniformly on $K_1$ with $\lambda=c_b$.

## B.8. $R_4$ (Gradient mismatch term)

Recall that $\nabla_\vartheta f(\vartheta,\tilde{\zeta}_1)-\nabla_\vartheta f(\vartheta,\zeta_1)$ is non-zero only when the latent variable $\zeta_1$ is updated, i.e., when

$$A_1 := \{1\in I_1^{(n_m)}\}$$

occurs. Therefore, we analyze

$$n_m^{\mathfrak{a}}\mathbb{E}\left[\langle\nabla_\vartheta f(\vartheta,\tilde{\zeta}_1)-\nabla_\vartheta f(\vartheta,\zeta_1),\Delta^{(n_m)}(\vartheta)\rangle\right] \tag{B.2}$$

by conditioning on $A_1$.

By the law of total expectation,

$$n_m^{\mathfrak{a}}\mathbb{E}\left[\langle\nabla_\vartheta f(\vartheta,\tilde{\zeta}_1)-\nabla_\vartheta f(\vartheta,\zeta_1),\Delta^{(n_m)}(\vartheta)\rangle\right]$$

$$= n_m^{\mathfrak{a}}\mathbb{P}(A_1)\mathbb{E}\left[\langle\nabla_\vartheta f(\vartheta,\tilde{\zeta}_1)-\nabla_\vartheta f(\vartheta,\zeta_1),\Delta^{(n_m)}(\vartheta)\rangle\,\Big|\,A_1\right]. \tag{B.3}$$

Since the mini-batch is sampled with replacement,

$$\mathbb{P}(A_1) = 1 - \left(1 - \frac{1}{n_m}\right)^{b^{(n_m)}}.$$

Note that $\Delta_\xi^{(n_m)}(\vartheta)$ and $\Delta_{\pi_0}^{(n_m)}(\vartheta)$ is independent of $A_1$, thus $\nabla_\vartheta f(\vartheta, \tilde{\zeta}_1) - \nabla_\vartheta f(\vartheta, \zeta_1)$ brings in a factor that is bounded by $n_m^{\mathfrak{b}-1}\|\nabla f\|_\infty$, keeping the effects from Gaussian noise and prior still inactive (As discussed in Section B.5) in the limit, thus we only consider $\Delta_\ell^{(n_m)}(\vartheta)$. Under $A_1$, the conditional expectation of the increment satisfies

$$\mathbb{E}[\Delta_\ell^{(n_m)}(\vartheta) \mid A_1] = \frac{c_h \, n_m^{-\mathfrak{h}+\mathfrak{w}}}{2}\left[\frac{1}{n_m\mathbb{P}(A_1)}\,\Gamma\nabla\ell\big(\hat{\theta}^{(n_m)} + (w^{(n_m)})^{-1}\vartheta; X_1, \tilde{\zeta}_1\big)\right.$$
$$\left.+ \left(1 - \frac{1}{n_m\mathbb{P}(A_1)}\right)\frac{1}{n_m - 1}\sum_{i=2}^{n_m}\Gamma\nabla\ell\big(\hat{\theta}^{(n_m)} + (w^{(n_m)})^{-1}\vartheta; X_i, \tilde{\zeta}_i\big)\right]. \qquad \text{(B.4)}$$

Substituting (B.4) into (B.3) yields an explicit decomposition of the gradient mismatch term.

Assume $\mathfrak{b} - 1 < 0$, so that $b^{(n_m)}/n_m \to 0$. Then

$$\mathbb{P}(A_1) = 1 - \left(1 - \frac{1}{n_m}\right)^{b^{(n_m)}} = \frac{b^{(n_m)}}{n_m} + O\left(\frac{b^{(n_m)2}}{n_m^2}\right).$$

Moreover,

$$\frac{1}{n_m\mathbb{P}(A_1)} = \frac{1}{b^{(n_m)}} + o\left(\frac{1}{b^{(n_m)}}\right).$$

Assuming $\mathfrak{a} = \mathfrak{h}$, the dominant contribution becomes

$$n_m^{\mathfrak{a}}\,\mathbb{E}\left[\langle\nabla_\vartheta f(\vartheta, \tilde{\zeta}_1) - \nabla_\vartheta f(\vartheta, \zeta_1), \Delta^{(n_m)}(\vartheta)\rangle\right]$$
$$= \frac{c_h\Gamma}{2}\,n_m^{\mathfrak{w}-1}\mathbb{E}_{\tilde{\zeta}}\left[\langle\nabla_\vartheta f(\vartheta, \tilde{\zeta}_1) - \nabla_\vartheta f(\vartheta, \zeta_1), \nabla\ell\big(\hat{\theta}^{(n_m)} + (w^{(n_m)})^{-1}\vartheta; X_1, \tilde{\zeta}_1\big)\rangle\right]$$
$$+ \frac{c_h\Gamma}{2}\,n_m^{\mathfrak{w}-1}\left\langle\mathbb{E}_{\tilde{\zeta}}\left[\nabla_\vartheta f(\vartheta, \tilde{\zeta}_1) - \nabla_\vartheta f(\vartheta, \zeta_1)\right], \frac{b^{(n_m)} - 1}{n_m - 1}\sum_{i=2}^{n_m}\nabla\ell\big(\hat{\theta}^{(n_m)} + (w^{(n_m)})^{-1}\vartheta; X_i\big)\right\rangle.$$

Note that

$$\frac{1}{n_m}\sum_{i=2}^{n_m}\nabla\ell\big(\hat{\theta}^{(n_m)} + (w^{(n_m)})^{-1}\vartheta; X_i\big) = \frac{1}{n_m}\sum_{i=1}^{n_m}\nabla\ell\big(\hat{\theta}^{(n_m)} + (w^{(n_m)})^{-1}\vartheta; X_i\big) - \frac{1}{n_m}\nabla\ell\big(\hat{\theta}^{(n_m)} + (w^{(n_m)})^{-1}\vartheta; X_1\big)$$
$$= n_m^{-\mathfrak{w}}(J_* \vartheta + R_J) - n_m^{-1}\nabla\ell\big(\hat{\theta}^{(n_m)} + (w^{(n_m)})^{-1}\vartheta; X_1\big),$$

where $R_J$ vanishes uniformly on $K_1$ as $n_m \to \infty$. So only consider the dominant terms,

$$\text{(B.2)} \lesssim n_m^{\mathfrak{w}-1}\|\nabla f\|_\infty\mathbb{E}_{\tilde{\zeta}}\|\nabla\ell\big(\hat{\theta}^{(n_m)} + (w^{(n_m)})^{-1}\vartheta; X_1, \tilde{\zeta}_1\big)\|$$
$$+ \|\nabla f\|_\infty\left(n_m^{\mathfrak{b}-1}J_\star\vartheta + n_m^{\mathfrak{b}+\mathfrak{w}-2}\nabla\ell\big(\hat{\theta}^{(n_m)} + (w^{(n_m)})^{-1}\vartheta; X_1\big)\right)$$

Since $\mathfrak{w} - 1 < 0$ and $\mathfrak{b} - 1 < 0$, and for $m$ large enough, $\max_{1 \le i \le n_m}\|\nabla\ell(\theta^*; X_i, \cdot)\| \le n_m^{1/p_2}$, all terms in (B.2) vanish, $R_4$ vanishes uniformly on $K_1$.

**B.9.** $R_5$ **(Hessian mismatch term)**

Let

$$A_1 := \{1 \in I_1^{(n_m)}\}, \qquad \mathbb{P}(A_1) = 1 - \left(1 - \frac{1}{n_m}\right)^{b^{(n_m)}}.$$

Again we will focus on

$$n_m^{\mathfrak{a}} \, \mathbb{E}\Big[(\Delta_\ell^{(n_m)}(\vartheta))^{\otimes 2} : \big(\nabla_{\vartheta\vartheta} f(\vartheta, \tilde{\zeta}_1) - \nabla_{\vartheta\vartheta} f(\vartheta, \zeta_1)\big)\Big]$$

$$= \mathbb{P}(A_1) \, n_m^{\mathfrak{a}} \, \mathbb{E}\Big[(\Delta_\ell^{(n_m)}(\vartheta))^{\otimes 2} : \big(\nabla_{\vartheta\vartheta} f(\vartheta, \tilde{\zeta}_1) - \nabla_{\vartheta\vartheta} f(\vartheta, \zeta_1)\big) \,\Big|\, A_1\Big].$$

Under the event $A_1 = \{1 \in I_1^{(n_m)}\}$, the conditional second moment of the likelihood increment admits the decomposition

$$\mathbb{E}\Big[(\Delta_\ell^{(n_m)}(\vartheta))^{\otimes 2} \mid A_1\Big]$$

$$= \left(\frac{c_h \, n_m^{-\mathfrak{h}+\mathfrak{w}}}{2}\right)^2 \Gamma\Bigg\{\frac{1}{b^{(n_m)}}\Bigg[\frac{\nabla\ell(\hat{\theta}^{(n_m)} + (w^{(n_m)})^{-1}\vartheta; X_1, \tilde{\zeta}_1)^{\otimes 2}}{n_m \, \mathbb{P}(A_1)}$$

$$+ \left(1 - \frac{1}{n_m \, \mathbb{P}(A_1)}\right) \frac{1}{n_m - 1} \sum_{i=2}^{n_m} \nabla\ell(\hat{\theta}^{(n_m)} + (w^{(n_m)})^{-1}\vartheta; X_i, \tilde{\zeta}_i)^{\otimes 2}\Bigg]$$

$$+ \frac{(b^{(n_m)} - 1)}{b^{(n_m)}}\Bigg[\left(\frac{1}{n_m \, \mathbb{P}(A_1)}\right)^2 \nabla\ell(\hat{\theta}^{(n_m)} + (w^{(n_m)})^{-1}\vartheta; X_1, \tilde{\zeta}_1)^{\otimes 2}$$

$$+ \frac{2}{n_m \, \mathbb{P}(A_1)}\left(1 - \frac{1}{n_m \, \mathbb{P}(A_1)}\right) \nabla\ell(\hat{\theta}^{(n_m)} + (w^{(n_m)})^{-1}\vartheta; X_1, \tilde{\zeta}_1) \otimes \left(\frac{1}{n_m - 1} \sum_{i=2}^{n_m} \nabla\ell(\hat{\theta}^{(n_m)} + (w^{(n_m)})^{-1}\vartheta; X_i, \tilde{\zeta}_i)\right)$$

$$+ \left(1 - \frac{1}{n_m \, \mathbb{P}(A_1)}\right)^2 \left(\frac{1}{n_m - 1} \sum_{i=2}^{n_m} \nabla\ell(\hat{\theta}^{(n_m)} + (w^{(n_m)})^{-1}\vartheta; X_i, \tilde{\zeta}_i)\right)^{\otimes 2}\Bigg]\Bigg\}\Gamma'.$$

Assuming $\mathfrak{a} = \mathfrak{h}$, the dominant contribution of

$$n_m^{\mathfrak{a}} \, \mathbb{E}\Big[(\Delta_\ell^{(n_m)}(\vartheta))^{\otimes 2} : \big(\nabla_{\vartheta\vartheta} f(\vartheta, \tilde{\zeta}_1) - \nabla_{\vartheta\vartheta} f(\vartheta, \zeta_1)\big)\Big]$$

becomes

$$\frac{c_h^2 n_m^{-\mathfrak{h}+2\mathfrak{w}+\mathfrak{b}-1}}{4}\Gamma\Bigg\{\frac{1}{b^{(n_m)}}\Bigg[\mathbb{E}_{\tilde{\zeta}}\Big[\frac{\nabla\ell(\hat{\theta}^{(n_m)} + (w^{(n_m)})^{-1}\vartheta; X_1, \tilde{\zeta}_1)^{\otimes 2}}{b^{(n_m)}} : \big(\nabla_{\vartheta\vartheta} f(\vartheta, \tilde{\zeta}_1) - \nabla_{\vartheta\vartheta} f(\vartheta, \zeta_1)\big)\Big] \qquad \text{(B.5)}$$

$$+ \left(\frac{b^{(n_m)} - 1}{b^{(n_m)}}\right) \frac{1}{n_m - 1} \sum_{i=2}^{n_m} \mathbb{E}_{\tilde{\zeta}}[\nabla\ell(\hat{\theta}^{(n_m)} + (w^{(n_m)})^{-1}\vartheta; X_i, \tilde{\zeta}_i)^{\otimes 2}] : \mathbb{E}_{\tilde{\zeta}}[\big(\nabla_{\vartheta\vartheta} f(\vartheta, \tilde{\zeta}_1) - \nabla_{\vartheta\vartheta} f(\vartheta, \zeta_1)\big)]\Bigg]$$

$$+ \frac{(b^{(n_m)} - 1)}{b^{(n_m)}}\Bigg[\frac{1}{b^{(n_m)2}}\mathbb{E}_{\tilde{\zeta}}[\nabla\ell(\hat{\theta}^{(n_m)} + (w^{(n_m)})^{-1}\vartheta; X_1, \tilde{\zeta}_1)^{\otimes 2} : \big(\nabla_{\vartheta\vartheta} f(\vartheta, \tilde{\zeta}_1) - \nabla_{\vartheta\vartheta} f(\vartheta, \zeta_1)\big)]$$

$$+ \frac{2(b^{(n_m)} - 1)}{b^{(n_m)2}}\mathbb{E}_{\tilde{\zeta}}[\nabla\ell(\hat{\theta}^{(n_m)} + (w^{(n_m)})^{-1}\vartheta; X_1, \tilde{\zeta}_1) : \big(\nabla_{\vartheta\vartheta} f(\vartheta, \tilde{\zeta}_1) - \nabla_{\vartheta\vartheta} f(\vartheta, \zeta_1)\big)]$$

$$\otimes \mathbb{E}_{\tilde{\zeta}}[\frac{1}{n_m - 1} \sum_{i=2}^{n_m} \nabla\ell(\hat{\theta}^{(n_m)} + (w^{(n_m)})^{-1}\vartheta; X_i, \tilde{\zeta}_i)]$$

$$+ \left(\frac{b^{(n_m)} - 1}{b^{(n_m)}}\right)^2 \mathbb{E}_{\tilde{\zeta}}\left(\frac{1}{n_m - 1} \sum_{i=2}^{n_m} \nabla\ell(\hat{\theta}^{(n_m)} + (w^{(n_m)})^{-1}\vartheta; X_i, \tilde{\zeta}_i)\right)^{\otimes 2} : \mathbb{E}_{\tilde{\zeta}}[\big(\nabla_{\vartheta\vartheta} f(\vartheta, \tilde{\zeta}_1) - \nabla_{\vartheta\vartheta} f(\vartheta, \zeta_1)\big)]\Bigg]\Bigg\}\Gamma'.$$

Then we want to write the terms with index $2, \cdots, n_m$ as summation over $1, \cdots, n_m$ and terms with index 1.

$$\frac{1}{n_m - 1} \sum_{i=2}^{n_m} \mathbb{E}_{\tilde{\zeta}}[\nabla \ell(\hat{\theta}^{(n_m)} + (w^{(n_m)})^{-1}\vartheta; X_i, \tilde{\zeta}_i)^{\otimes 2}]$$
$$= \frac{n_m}{n_m - 1}\left(\tilde{I}_\star + R_I - n_m^{-1}\mathbb{E}_{\tilde{\zeta}}\nabla \ell(\hat{\theta}^{(n_m)} + (w^{(n_m)})^{-1}\vartheta; X_1, \tilde{\zeta}_1)^{\otimes 2}\right),$$

$$\mathbb{E}_{\tilde{\zeta}}[\frac{1}{n_m - 1} \sum_{i=2}^{n_m} \nabla \ell(\hat{\theta}^{(n_m)} + (w^{(n_m)})^{-1}\vartheta; X_i, \tilde{\zeta}_i)]$$
$$= \frac{n_m}{n_m - 1}\left(n_m^{-\mathfrak{w}}(J_\star \vartheta + R_J) - n_m^{-1}\nabla \ell(\hat{\theta}^{(n_m)} + (w^{(n_m)})^{-1}\vartheta; X_1)\right)$$

$$\mathbb{E}_{\tilde{\zeta}}\left(\frac{1}{n_m - 1}\sum_{i=2}^{n_m} \nabla \ell(\hat{\theta}^{(n_m)} + (w^{(n_m)})^{-1}\vartheta; X_i, \tilde{\zeta}_i)\right)^{\otimes 2}$$
$$= \frac{1}{(n_m - 1)^2}\Bigg\{\left(\sum_{i=1}^{n_m} \nabla \ell(\hat{\theta}^{(n_m)} + (w^{(n_m)})^{-1}\vartheta; X_i)\right)^{\otimes 2} + \sum_{i=1}^{n_m} \mathbb{E}_{\tilde{\zeta}}[\nabla \ell(\hat{\theta}^{(n_m)} + (w^{(n_m)})^{-1}\vartheta; X_i, \tilde{\zeta}_i)^{\otimes 2}]$$
$$- \sum_{i=1}^{n_m} \nabla \ell(\hat{\theta}^{(n_m)} + (w^{(n_m)})^{-1}\vartheta; X_i)^{\otimes 2} - 2\nabla \ell(\hat{\theta}^{(n_m)} + (w^{(n_m)})^{-1}\vartheta; X_1)\left(\sum_{i=1}^{n_m} \nabla \ell(\hat{\theta}^{(n_m)} + (w^{(n_m)})^{-1}\vartheta; X_i)\right)$$
$$+ 2\nabla \ell(\hat{\theta}^{(n_m)} + (w^{(n_m)})^{-1}\vartheta; X_1)^{\otimes 2} - \mathbb{E}_{\tilde{\zeta}}[\nabla \ell(\hat{\theta}^{(n_m)} + (w^{(n_m)})^{-1}\vartheta; X_1, \tilde{\zeta}_1)^{\otimes 2}]\Bigg\}$$

So only consider the dominant terms,

$$(\text{B.5}) \lesssim n_m^{-1}\mathbb{E}_{\tilde{\zeta}}[\nabla \ell(\hat{\theta}^{(n_m)} + (w^{(n_m)})^{-1}\vartheta; X_1, \tilde{\zeta}_1)^{\otimes 2} : (\nabla_{\vartheta\vartheta} f(\vartheta, \tilde{\zeta}_1) - \nabla_{\vartheta\vartheta} f(\vartheta, \zeta_1))] + n_m^{\mathfrak{b}-1}\|\nabla^{\otimes 2}\|_\infty \tilde{I}_\star$$
$$+ n_m^{\mathfrak{b}-1-\mathfrak{w}}\mathbb{E}_{\tilde{\zeta}}[\nabla \ell(\hat{\theta}^{(n_m)} + (w^{(n_m)})^{-1}\vartheta; X_1, \tilde{\zeta}_1) \otimes J_\star \vartheta : (\nabla_{\vartheta\vartheta} f(\vartheta, \tilde{\zeta}_1) - \nabla_{\vartheta\vartheta} f(\vartheta, \zeta_1))]$$
$$+ \|\nabla^{\otimes 2}\|_\infty\left(n_m^{\mathfrak{b}-\mathfrak{h}-1}(J_\star\vartheta)^{\otimes 2} + n_m^{2\mathfrak{b}-2}(\tilde{I}_\star - I_\star) + n_m^{2\mathfrak{b}-2-\mathfrak{w}}J_\star \vartheta \otimes \nabla \ell(\hat{\theta}^{(n_m)} + (w^{(n_m)})^{-1}\vartheta; X_1)\right)$$
$$\lesssim \|\nabla^{\otimes 2}\|_\infty\Bigg\{n_m^{-1}\mathbb{E}_{\tilde{\zeta}}\|\nabla \ell(\hat{\theta}^{(n_m)} + (w^{(n_m)})^{-1}\vartheta; X_1, \tilde{\zeta}_1)\|_\infty^2 + n_m^{\mathfrak{b}-1}\tilde{I}_\star + n_m^{\mathfrak{b}-1-\mathfrak{w}}\mathbb{E}_{\tilde{\zeta}}\|\nabla \ell(\hat{\theta}^{(n_m)} + (w^{(n_m)})^{-1}\vartheta; X_1, \tilde{\zeta}_1)\|$$
$$+ n_m^{\mathfrak{b}-\mathfrak{h}-1}(J_\star\vartheta)^{\otimes 2} + n_m^{2\mathfrak{b}-2}(\tilde{I}_\star - I_\star) + n_m^{2\mathfrak{b}-2-\mathfrak{w}}J_\star \vartheta \otimes \nabla \ell(\hat{\theta}^{(n_m)} + (w^{(n_m)})^{-1}\vartheta; X_1)\Bigg\}$$

Since $\mathfrak{w} - 1 < 0$ and $\mathfrak{b} - 1 < 0$, and for $m$ large enough, $\max_{1 \le i \le n_m}\|\nabla\ell(\theta^*; X_i, \cdot)\| \le n_m^{1/p_2}$, $R_5$ vanishes uniformly on $K_1$.

## B.10. $R_6$ (third-order remainder term)

By the triangle inequality,

$$n_m^{\mathfrak{a}}\,\mathbb{E}\left[\frac{1}{6}\,\nabla_\vartheta^{\otimes 3} f(\vartheta + S\Delta^{(n_m)}(\vartheta), \tilde{\zeta}_1)(\Delta^{(n_m)}(\vartheta), \Delta^{(n_m)}(\vartheta), \Delta^{(n_m)}(\vartheta))\right]$$
$$\le \frac{27}{6}\,n_m^{\mathfrak{a}}\,\|\nabla_\vartheta^{\otimes 3} f\|_\infty\left(\mathbb{E}\|\Delta_\xi^{(n_m)}(\vartheta)\|^3 + \mathbb{E}\|\Delta_{\pi_0}^{(n_m)}(\vartheta)\|^3 + \mathbb{E}\|\Delta_\ell^{(n_m)}(\vartheta)\|^3\right).$$

Moreover,

$$\mathbb{E}\|\Delta_\xi^{(n_m)}(\vartheta)\|^3 \le n_m^{-\frac{3}{2}(\mathfrak{h}+\mathfrak{t}-2\mathfrak{w})}\left(\frac{c_h}{2c_\beta}\|\Gamma\|^{3/2}\,2^{3/2}\,\frac{\Gamma\left(\frac{d+3}{2}\right)}{\Gamma\left(\frac{d}{2}\right)}\right), \qquad \mathfrak{a} - \frac{3}{2}(\mathfrak{h} + \mathfrak{t} - 2\mathfrak{w}) < 0.$$

Also,

$$\mathbb{E}\|\Delta_{\pi_0}^{(n_m)}(\vartheta)\|^3 \leq \left(\frac{c_h}{2} n_m^{-\mathfrak{h}+\mathfrak{w}-1} \|\Gamma\|\right)^3 \left(\|\nabla \log \pi_0(\theta^*)\| + L_0\|\hat{\theta}^{(n_m)} - \theta^*\| + \frac{L_0(2R_0 + 2c_0)}{n_m^{\mathfrak{w}}}\right)^3, \qquad \mathfrak{a} - 3\mathfrak{h} + 3\mathfrak{w} - 3 < 0.$$

Finally,

$$\mathbb{E}\|\Delta_{\ell}^{(n_m)}(\vartheta)\|^3 \leq \left(\frac{c_h\|\Gamma\|}{2}\right)^3 \left(n_m^{1/p_2 - \mathfrak{h} + \mathfrak{w}} + n_m^{1/p_3 - \mathfrak{h} + \mathfrak{w}} \Phi^{(n_m)} + n_m^{1/p_3 - \mathfrak{h}}\right)^3.$$

Therefore, $R_6$ vanishes uniformly.

## C. Experimental Details

This appendix provides additional details on the experimental setups used in Section 4, including the synthetic data-generating distributions and real-data description. Code to reproduce all experiments is available at `https://github.com/shawngyn-stack/LVM_scaling_limits`.

### C.1. Gaussian Mixture Model (GMM) Experiments

We consider a Gaussian mixture model with $K$ components in $\mathbb{R}^d$ and dataset size $N$. For each observation $i \in \{1, \dots, N\}$, a latent cluster label is drawn as

$$z_i \sim \text{Categorical}(\pi), \qquad z_i \in \{1, \dots, K\},$$

and the observation is generated according to

$$X_i \mid z_i = k \sim \mathcal{N}\big(\mu_k, \text{diag}(\sigma_k^2)\big),$$

where $\mu_k \in \mathbb{R}^d$ denotes the component mean and $\sigma_k \in \mathbb{R}_+^d$ parameterizes a diagonal covariance matrix. The global parameters are $\{\mu_k, \sigma_k\}_{k=1}^K$, while the latent variables are the cluster assignments $\{z_i\}_{i=1}^N$.

### C.2. Latent Dirichlet Allocation (LDA)

We consider the standard LDA generative model with $K$ topics and a vocabulary of size $V$. Let $D$ denote the number of documents. For each topic $k \in \{1, \dots, K\}$, we draw a topic-word distribution

$$\beta_k \sim \text{Dirichlet}(\beta \mathbf{1}_V), \qquad \beta_k \in \Delta^{V-1},$$

independently across $k$. Here $\beta > 0$ is a symmetric Dirichlet hyperparameter and $\mathbf{1}_V$ denotes the all-ones vector in $\mathbb{R}^V$.

For each document $d \in \{1, \dots, D\}$, we draw a document-topic proportion vector

$$\theta_d \sim \text{Dirichlet}(\alpha \mathbf{1}_K), \qquad \theta_d \in \Delta^{K-1},$$

independently across documents, where $\alpha > 0$ is a symmetric concentration parameter. We then sample the document length $L_d$ independently from a discrete uniform distribution over a fixed range. Conditional on $\theta_d$, each token $n \in \{1, \dots, L_d\}$ is generated by first sampling a topic assignment

$$z_{dn} \mid \theta_d \sim \text{Categorical}(\theta_d),$$

and then sampling the observed word

$$w_{dn} \mid z_{dn} = k \sim \text{Categorical}(\beta_k).$$

We collect each document as a sequence of word indices $w_{d,1:L_d} \in \{1, \dots, V\}^{L_d}$.

In our experiments, the global parameters correspond to the topic-word distributions $\{\beta_k\}_{k=1}^K$ (equivalently, the matrix $\beta \in \mathbb{R}^{K \times V}$ with rows on the simplex), while the latent variables are the token-level topic assignments $\{z_{dn}\}$. In the experiments, the document-topic proportions $\{\theta_d\}$ are integrated out, and Gibbs updates are performed over the token-level assignments $\{z_{dn}\}$. For further details, we refer the reader to Patterson & Teh (2013).

## C.3. Real-World Datasets

**Flow cytometry (GMM).** We use a real flow cytometry dataset in which each observation corresponds to a single cell. As input features, we retain only the four fluorescence intensity channels FL1.H–FL4.H, which measure marker-specific protein expression levels. All features are standardized prior to model fitting. Ground-truth cell population labels are available and are used for evaluation.

**20 Newsgroups (LDA).** For topic-modeling experiments, we use the 20 Newsgroups dataset as provided by `scikit-learn` (Pedregosa et al., 2011). Documents are represented as bags of words after standard preprocessing.

## C.4. Setup of SVI and computational cost

We used the default schedule recommended by Hoffman et al. (2013) and verified it converges; results are robust to moderate changes in $\tau_0$ and $\kappa$. The hyperparameters' values are as follows.

GMM: mean-field Normal–Gamma variational family initialized via k-means; minibatch size 256; prior parameters $\alpha_0 = 1.0$, $\beta_0 = 1.0$, $a_0 = 2.0$, $b_0 = 2.0$; Robbins–Monro step-size schedule $\rho_t = (\tau_0 + t)^{-\kappa}$ with $\tau_0 = 10$ and $\kappa = 0.7$.

LDA (synthetic and real data): semi-collapsed SVI with variational family $q(\pi) = \prod_k \text{Dirichlet}(\lambda_k)$ and categorical local factors; minibatch size 64; 10 local coordinate-ascent updates per document; symmetric priors $\alpha = \beta = 0.1$; same Robbins–Monro schedule.

In our synthetic GMM experiments, SVI typically stabilizes within roughly 20 iterations, whereas SGLD–Gibbs requires an initial burn-in period of about 200 iterations, followed by additional iterations used for posterior sampling. On a per-iteration basis, the cost of SGLD–Gibbs is approximately 0.3–0.6× that of SVI when $S = 1$, and approximately 1.6–3.0× when $S = 5$. Consequently, the overall cost of SGLD–Gibbs depends on the number of retained posterior samples. In our experiments, the total cost needed to obtain stable SGLD–Gibbs estimates was roughly 5–10× that of SVI. This additional cost was associated with accuracy improvements of about 20–50%, together with better calibrated uncertainty quantification.

