# OpenReview forum: "Large-scale Uncertainty Quantification for Latent Variable Models Using Subsampling Markov Chain Monte Carlo"
_ICML.cc/2026/Conference — ICML 2026 regular_

### Official Review · Reviewer_YFe5 · 2026-03-08

**Soundness:** 2
**Presentation:** 2
**Significance:** 3
**Originality:** 3
**Overall Recommendation:** 4
**Confidence:** 2

**Summary:**

The authors provide a theoretical analysis of SGLD-Gibbs for local latent variable models. They derive
a joint asymptotic limit for the global/local latent variables in which the global latent variables converge to a Ornstein–Uhlenbeck process and the local latent variables follow a Poisson-driven Gibbs jump process. This theoretical characterization is used to derive guidance for tuning the SGLD step size and temperature. The empirical experiments are used to verify the derived scaling limit. In addition the authors compare the uncertainty quantification of SGLD-Gibbs to SVI (stochastic variational inference) using both synthetic and real-world data.

**Compliance With Llm Reviewing Policy:**

Affirmed.

**Final Justification:**

While I would not champion this submission for acceptance--in large part due to the somewhat limited empirical validation--in the balance I believe it could be a good contribution to the ICML community, and as such I recommend acceptance. As I mentioned in my review, I believe it represents an interesting case of a sophisticated theoretical analysis that provides some useful practical guidance w.r.t. tuning schemes, something that can be hard to come by for complex algorithms like SGLD-Gibbs.

**Key Questions For Authors:**

-  What is the relationship of this work to "Authors, A. Accurate large-scale uncertainty quantification using stochastic gradient Markov chain Monte Carlo"?
- What are the computational requirements of SVI vs SGLD-Gibbs in your experiments? If it is the case that, for example, SGLD-Gibbs is much more computationally heavy, then it is unfair/unhelpful to leave out this information. Similarly, without any details on how you do SVI it is impossible to assess whether or not you're following best practices. As such it is difficult to assess the implications of your comparison to SVI.
- Re: "{1, . . . , n} (with replacement)" in Sec 2.2; please clarify if this is merely for theoretical convenience or if there is something deeper in play.
- I think the reader could benefit from more discussion of why/when one might choose "Bernstein–von Mises" vs "Bagged posterior."

**Limitations:**

yes

**Strengths And Weaknesses:**

**Soundness:**

As far as I can tell the submission is technically sound, although I emphasize that I have not gone through the proofs nor am I very familiar with the relevant asymptotic theory. The experiments provide too few details for me to make an assessment as to whether SGLD-Gibbs in general, and the author's tuning scheme in particular, robustly outperforms SVI. An ablation experiment that compares to a simpler tuning scheme could help in this regard.

**Presentation:**

The paper is generally easy to follow, although I'm missing some important details. In particular there are essentially zero details on how SVI is implemented in the experiments or indeed what particular SVI scheme is being used. If the authors want to make strong empirical claims in this regard, they should probably expand the set of variational inference baselines being used. It would appear to me to be somewhat unfair to rely solely on non-black-box SVI methods that rely on conjugacy, while allowing allowing SGLD to leverage black-box gradients w.r.t. $\Theta$.

**Significance:**

The submission addresses a relatively general algorithm---SGLD-Gibbs---that is applicable to a relatively general class of models---local latent variable models that admit Gibbs update for local latent variables. As such it should be of interest to a number of ICML subcommunities. I believe it represents an interesting case of a seemingly sophisticated theoretical analysis that provides some useful practical guidance w.r.t. tuning schemes.

**Originality:**

To the best of my knowledge the submission is novel, although I am not very familiar with the relevant literature.

---

> ### Author Rebuttal · Authors · 2026-03-31
>
> We thank the reviewer for their careful review and for highlighting the novelty of our theoretical analysis and its potential practical guidance. We appreciate the constructive comments and questions, which we address below.
>
> Q1.
> Our submission “Accurate large-scale uncertainty quantification using stochastic gradient Markov chain Monte Carlo” addresses the limitations of continuous-time approximations (such as the one we use in the current paper) by instead using a discrete-time approximation with linearized gradients. Our results in that paper are relevant, for example, when using very large batch sizes that require a large step size to obtain the correct stationary distribution. The results in that paper are limited “classical” settings where the limiting sample paths follow an OU process. Hence, our other submission complements our SGLD-Gibbs analysis. For example, a natural direction for future work is to combine the ideas from the two papers to develop a discrete-time approximation for SGLD-Gibbs that would be more accurate in the large-batch regime.
>
> Q2.
> We discussed the computational cost and hyperparameter tuning in answers to Q4 of reviewer 1 and Q5 of reviewer 3. In short, we used the default schedule recommended by Hoffman et al. (2013) and verified it converges; results are robust to moderate changes. SGLD-Gibbs cost between 5-10 times as much as SVI, but the accuracy was improved by a factor of 20%-50% in different tasks and provided better uncertainty quantification. We will add these details in the camera-ready version.
>
>
> Q3.
> The use of sampling with replacement in Section 2.2 is primarily for theoretical convenience. Notably, Negrea et al. (2022) provide results for sampling without replacement as well; we therefore expect our conclusions to remain consistent under that scheme. The only differences would be constant-level deviations occurring when the batch size $b$ is of order $n$. We will address this in more detail in the revision.
>
> Q4.
> Thank you for raising this point regarding the choice between the Bernstein–von Mises approximation and the Bagged posterior. The two targets correspond to different inferential goals. The BvM covariance $J_\star^{−1}$​ is the classical posterior target under correct model specification, while the bagged-posterior covariance $S_\star=J_\star^{-1}I_\star J_\star^{-1}$ takes the sandwich form and is robust to misspecification. In practice, one would prefer BvM when the model is believed to be well-specified, and the bagged posterior when robustness to misspecification is a concern. We will add this discussion to the paper.

---

> > ### Author Rebuttal · Reviewer_YFe5 · 2026-03-31
> >
> > I thank the authors for their response. I keep my score as is, as I would want to see a more thorough-going empirical evaluation in order to raise my score.

---

### Official Review · Reviewer_uFwH · 2026-03-09

**Soundness:** 3
**Presentation:** 3
**Significance:** 3
**Originality:** 3
**Overall Recommendation:** 5
**Confidence:** 3

**Summary:**

A scalable approach of approximate Bayesian inference for latent variable models is to perform subsampled SGLD for the global parameter and Gibbs sampling for the latent variables. An existing work (Negrea et al.) provides a theoretical guidance of choosing hyperparameters for SGLD on models without latent variables. This work generalizes the existing theoretical results to the SGLD-Gibbs setting:
- The new theoretical results are very similar to the existing ones after considering the algorithm-induced uncertainty from the latent variables. The additional uncertainty could be reduced by increasing the size of Gibbs samples.
- In the limit, the global parameter and any local parameter are independent. The distributions of them depend on the hyperparameters for the step size and the batch size.
- By properly setting the hyperparameters, we can choose the limiting distribution of the global parameter to have the Bernstein–von Mises covariance or the bagged posterior covariance, accounting for both well-specified and mis-specified cases.
- The experiments show that under the guidance of the work, SGLD–Gibbs achieves better-calibrated posterior uncertainty and better predictive performance than stochastic variational inference.

**Compliance With Llm Reviewing Policy:**

Affirmed.

**Final Justification:**

I like the paper and remain positive.

**Key Questions For Authors:**

I have a few comments and questions:
- In Section 2.2, $\otimes$ is used, but the definition is in the prior work. I believe it should be clearly stated here.
- Section 2.4, after Proposition 2.2: "Theorem 2.2" should be changed to "Proposition 2.2".
- There are two problems in Table 1. First, I believe $w_1$ and $w_2$ should be swapped. Second, by setting $w_1=w_2=1$, the Bagged post. covariance does not recover the BvM covariance. What am I missing here?
- The results of Negrea et al. include mixing time for different algorithms. However, it is only briefly mentioned after Corollary 3.2 in this work. Should I assume that the $1/\lambda$ rate applies to all cases?
- How are the hyperparameters for SVI chosen?
- Are there additional experimental results with naive hyperparameter choosing or on a larger-scale dataset?

**Limitations:**

I believe the impact statement of the work is missing, although I do not see any negative social impact of this work. The authors have adequately discussed limitations in Section 5.

**Strengths And Weaknesses:**

Strengths:
- The generalization of existing theoretical results to latent variable models is natural. It is interesting to see there is a set of hyperparameters that is good jointly for both the global and local parameters. And the limiting results considering multiple Gibbs samples connect this work with the prior work.
- The presentation is clear. The main theoretical results and the proof sketch are well presented in the main text. The guidance for practitioners to choose hyperparameters is summarized as a table.

Weaknesses:
- The experiments compare SGLD-Gibbs (with the chosen hyperparameters) against SVI, which may not be fair. There are no details on how the hyperparameters of SVI are chosen.
- The core contribution of this work is the guidance of choosing hyperparameters. Thus, an important baseline is SGLD where the hyperparameters are chosen in another way (e.g., cross-validation). The current results seem to be claiming that SGLD+Gibbs is better than SVI on latent variable models, which may be correct but do not perfectly align with the main contribution.
- The title has the keyword "large-scale" but the experiments are conducted on relatively small datasets. This is not a major weakness for a theoretical work, but conducting larger experiments and reporting hardware specifications will greatly strengthen this work.

---

> ### Author Rebuttal · Authors · 2026-03-31
>
> We thank the reviewer for their careful reading and positive remarks on the clarity of presentation and the naturalness of the generalization. We appreciate the constructive feedback on the experiments and the reviewer’s careful identification of several small writing issues in the current draft, which we will correct in the revision.
>
> Q1.
> Thank you for catching this.  For vectors $\(a,b \in \mathbb{R}^d\)$, define the outer product $\(a \otimes b \in \mathbb{R}^{d \times d}\)$ given by  $\((a \otimes b)_{ij} = a_i b_j\)$, and write $\(a^{\otimes 2} := a \otimes a\)$.  And $\(\nabla \otimes \nabla = \nabla^{\otimes 2}\)$ denotes the Hessian operator. We will add this in Section 2.2 in the revision.
>
> Q2.
> Thank you; this is correct. “Theorem 2.2” after Proposition 2.2 should indeed read “Proposition 2.2.”  We will change this in the revision.
>
> Q3.
> Thank you for pointing this out. We will unify the use of $w_1$ and $w_2$ in the revision. Regarding the second point, the bagged-posterior row references SGLD-FP rather than SGLD, which causes the inconsistency: setting $w_1 = w_2 =1$ does not recover the BvM covariance under SGLD. We will unify the table to use SGLD throughout and update the bagged-posterior row to the corresponding tuning $\beta = c_\beta n$ and $h=4b(1−c_\beta)/(nc_\beta)$ in the revision.
>
> Q4.
>  The 1/λ mixing rate in Corollary 3.2 applies only to the latent-variable jump process in the regime $h+b=1$. For the global parameter, the relevant mixing heuristic is inherited from Theorem 2.1 via the drift matrix B of the limiting Ornstein–Uhlenbeck process, as in Negrea et al. We will clarify this distinction in the revision.
>
> Q5.
> Thank you for asking. We used the default schedule recommended by Hoffman et al. (2013) and verified it converges; results are robust to moderate changes in $\tau_0$​ and $\kappa$. The hyperparameters’ values are as follows.
> GMM: mean-field Normal--Gamma variational family initialized via k-means; minibatch size $256$; prior parameters $\alpha_0=1.0$, $\beta_0=1.0$, $a_0=2.0$, $b_0=2.0$; Robbins--Monro step-size schedule $\rho_t = (\tau_0 + t)^{-\kappa}$ with $\tau_0=10$ and $\kappa=0.7$.
> LDA (synthetic and real data): semi-collapsed SVI with variational family $q(\pi)=\prod_{k}\mathrm{Dirichlet}(\lambda_k)$ and categorical local factors; minibatch size $64$; $10$ local coordinate-ascent updates per document; symmetric priors $\alpha=\beta=0.1$; same Robbins--Monro schedule.
>
>
> Q6.
> We tried simple “naive” hyperparameter setups, for example using a large stepsize with a small batch size. This led to very unstable convergence, unbalanced calibration and the sampled latent variables z did not match the predicted posteriors. A similar issue occurs if we don’t use a preconditioner.
> For dataset size, our GMM experiments used 50k data points, and the real-data LDA experiments involved 18,000 documents of average length 500. We don’t have additional experiments for larger dataset sizes. For synthetic data we believe the size is large enough to validate our theory empirically. For real data LDA, scaling to significantly larger datasets is computationally prohibitive within the revision period.

---

> > ### Author Rebuttal · Reviewer_uFwH · 2026-03-31
> >
> > Thank you for the responses. My concerns are mostly addressed so I will keep my positive stance.

---

### Official Review · Reviewer_idwX · 2026-03-12

**Soundness:** 3
**Presentation:** 4
**Significance:** 3
**Originality:** 3
**Overall Recommendation:** 4
**Confidence:** 3

**Summary:**

Stochastic gradient Langevin dynamics combined with Gibbs updates (SGLD-Gibbs) is widely used in Bayesian inference but lacks principled hyperparameter tuning strategies. This paper studies SGLD-Gibbs in latent variable models (LVMs) by analyzing the joint asymptotic behavior of the rescaled dynamics of both global and latent parameters. It shows that, under appropriate scaling, the rescaled global parameter dynamics converges to an Ornstein-Uhlenbeck process, while the latent variable dynamics converges to a pure-jump Markov process, and that the two limiting processes are independent. Based on this asymptotic analysis, the authors propose a hyperparameter tuning strategy. Empirical studies validate the theoretical predictions on synthetic data, and real-data experiments compare SGLD-Gibbs with stochastic variational inference (SVI).

**Compliance With Llm Reviewing Policy:**

Affirmed.

**Key Questions For Authors:**

- Q1. Focus of the paper

The paper touches on several related themes, including uncertainty quantification, hyperparameter tuning, and the asymptotic analysis of SGLD–Gibbs. However, the primary focus of the paper is not entirely clear. The title suggests an emphasis on uncertainty quantification, while the introduction highlights hyperparameter tuning motivated by theoretical analysis. In contrast, much of the paper focuses on establishing asymptotic results, and the experiments mainly aim to support these theoretical findings. Could the authors clarify which of these aspects represents the central objective of the paper? Clarifying this point would help improve the consistency between the stated motivation, the theoretical development, and the empirical evaluation.

- Q2. Comparison with other tuning strategies

The paper proposes hyperparameter tuning strategies for SGLD–Gibbs, but the empirical evaluation does not include comparisons with alternative tuning strategies within SGLD–Gibbs algorithms. Could the authors clarify how the proposed strategy performs relative to other commonly used tuning approaches? Such comparisons would help assess the practical effectiveness of the proposed strategy and could strengthen the empirical validation of the paper.

- Minor comments:
1. In Section 3.4, the notations $w_1$ and $w_2$ are used with two different meanings. On one hand, they are used as the constants in tuning of inverse temperature and step size, while in the asymptotic covariance they are used as the weight of the inverse Fisher matrix and sandwich matrix. Those two meanings simultaneously appear in Table 2, which may lead to confusion.
2. In the interpretation of Theorem 3.1, the article mentions that the freezing of latent variables results in the persistent bias of global parameters, which can be more clearly illustrated since the global parameter convergence does not require an additional scaling constraint.

**Limitations:**

yes

**Strengths And Weaknesses:**

- Soundness: This article is technically sound overall. The theoretical analysis is rigorous, and the development builds on established asymptotic analyses for stochastic gradient methods. The assumptions appear standard for latent variable models and stochastic gradient dynamics, and the overall proof strategy is clearly outlined. The simulation studies verify the asymptotic convergence and illustrate the role of the generalized Gibbs approximation under the proposed tuning strategy.
That said, some aspects of the empirical validation could be strengthened. In particular, the proposed design does not appear to be fully aligned with the ultimate goal of hyperparameter tuning. In the real-world dataset experiments, SGLD–Gibbs is compared with SVI across several tasks, but the proposed tuning strategy is not compared with alternative tuning strategies within SGLD–Gibbs algorithms.

- Presentation: This article is organized in a standard form for analyzing the asymptotic behavior of SGLD–Gibbs for latent variables, with the preliminaries and proof sketch clearly presented. The article clearly introduces the preliminary models and summarizes relevant results from prior work, properly positioning itself within the context of Bayesian inference. The basic ideas underlying the asymptotic theory can be readily extracted from the corresponding sections. In addition, the hyperparameter settings are clearly described, which facilitates reproducibility.

- Significance: This article extends the stochastic convergence analysis of SGLD to the convergence of SGLD–Gibbs in latent variable models, and the setting applies to a wide range of practical scenarios. The analysis contributes to a better understanding of the behavior of these sampling methods. In addition, the derived uncertainty quantification and hyperparameter tuning strategies appear straightforward to apply in the corresponding algorithms.

- Originality: The asymptotic analysis in this paper includes the convergence of both global and latent parameters, as well as their asymptotic independence. The convergence for the global parameters is derived in a manner similar to related work, with the additional noise introduced by Gibbs sampling incorporated into the framework. On the other hand, the convergence analysis for the latent variables and the asymptotic independence property are established through more delicate technical arguments. While the hyperparameter tuning strategies follow ideas similar to those in related work, the analysis provides a useful extension of existing theoretical results to the SGLD–Gibbs setting.

---

> ### Author Rebuttal · Authors · 2026-03-31
>
> We thank the reviewer for their careful review and for their positive assessment of the paper's technical soundness, clarity of presentation, and reproducibility. We appreciate the constructive questions and minor comments, which we address below.
>
> Q1.
> The central objective of the paper is to use asymptotic analysis to derive principled tuning guidance for SGLD–Gibbs that ensures meaningful uncertainty quantification (UQ) in latent variable models. The scaling-limit results are the main technical tool rather than the end goal. In other words, UQ motivates the work, our theoretical analysis provides the tuning principles to ensure meaningful UQ, and the experiments validate these theory-guided insights. We will revise the introduction to make this structure more explicit.
>
> Q2.
> To our knowledge, the closest alternative tuning strategy for SGMCMC is KSD-based tuning in Coullon et al. (2023) ‘Efficient and generalizable tuning strategies for stochastic gradient MCMC’. Our approach differs in nature: rather than optimizing an external discrepancy criterion, it is derived directly from the asymptotic analysis and targets scaling regimes that yield interpretable uncertainty quantification. We view the two approaches as complementary — our theory can restrict attention to principled scaling regimes, within which KSD-based criteria could provide further refinement. A direct empirical comparison would be an interesting direction which we will mention in our conclusion and leave for future work.
>
> Minor comments:
>
> Q1.
> Thank you for pointing this out. We will unify the use of $w_1$ and $w_2$ in the revision.
>
> Q2.
> We apologize for the note being more clear on this point.  The statement as written is imprecise: setting $h+b<1$ does not directly cause bias in the global parameters. Rather, in this regime, the latent-variable dynamics become degenerate on the macroscopic timescale; this is because $z_1$ is refreshed too infrequently relative to the evolution of $\theta$, so the latent-variable process effectively freezes and loses any meaningful limiting behavior. We will clarify these points in the camera-ready version.

---

> > ### Author Rebuttal · Reviewer_idwX · 2026-04-01
> >
> > Thanks for the responses. Comments are mainly on writing and presentation, which are hopefully addressed in the final version. I will keep the positive score based on the quality of the work.

---

### Official Review · Reviewer_Jxtd · 2026-03-13

**Soundness:** 3
**Presentation:** 3
**Significance:** 3
**Originality:** 4
**Overall Recommendation:** 4
**Confidence:** 3

**Summary:**

The paper studies uncertainty quantification for latent variable models under scalable subsampling MCMC, focusing on SGLD–Gibbs. It develops a joint scaling-limit theory in which the global parameters converge to a diffusion process while individual latent variables converge to jump processes, thereby characterizing how intermittent Gibbs updates introduce additional algorithm-induced uncertainty into the stationary distribution of the global parameters. Building on this analysis, the paper derives practical hyperparameter tuning rules for step size, minibatch size, inverse temperature, and the number of Gibbs samples, with the goal of achieving statistically meaningful uncertainty estimates. Experiments on synthetic Gaussian mixture models and LDA, as well as real-data evaluations on flow cytometry and 20 Newsgroups, suggest that the proposed tuning framework improves posterior calibration, parameter estimation, and predictive performance relative to stochastic variational inference.

**Compliance With Llm Reviewing Policy:**

Affirmed.

**Key Questions For Authors:**

Key Questions For Authors

1.How sensitive is the proposed tuning strategy to deviations from the ideal asymptotic scaling regime in finite-sample practice, especially when h + b is not exactly equal to 1?

2.Can the authors better disentangle the effect of the theory-guided tuning rules from the effect of using generalized Gibbs updates with larger S?

3.Which parts of the analysis fundamentally rely on the assumption of local latent variables with conditional independence, and how difficult would it be to extend the theory to models with shared or interacting latent structures?

4.Can the authors provide a clearer comparison of computational cost and runtime across SGLD-Gibbs with different S values and SVI?

**Limitations:**

Yes.

**Strengths And Weaknesses:**

Strengths

Originality:
The paper makes a notable theoretical contribution by extending scaling-limit analysis from standard stochastic-gradient methods to latent variable models. By jointly analyzing global parameters and local latent variables under space-time rescaling, the paper reveals a nontrivial jump-diffusion limit. In particular, the identification of an additional algorithm-induced uncertainty term, denoted as M_*, arising from latent-variable updates provides a fresh perspective on how Gibbs refreshes affect the stationary distribution of global parameters.

Significance:
The work addresses a practically relevant problem. Tuning SGLD is already difficult, and principled guidance for SGLD-Gibbs in latent variable models has been limited. A key strength of the paper is that it translates asymptotic theory into concrete hyperparameter prescriptions, summarized clearly in Table 2. This gives practitioners a more principled way to target meaningful uncertainty quantification, potentially improving over standard variational approximations.

Soundness:
The paper appears technically strong overall. The theoretical development clearly specifies the relevant scaling regimes and limiting processes, and the proof sketch is well structured. The empirical section is also aligned with the theoretical claims: synthetic experiments assess calibration in a setting with known ground truth, while real-data experiments on flow cytometry and 20 Newsgroups support the practical utility of the proposed tuning strategy relative to SVI.

Presentation:
The paper is well organized and generally clear. The progression from problem setup, to theory, to tuning implications, and finally to experiments is easy to follow. The side-by-side summary tables for standard SGLD and SGLD-Gibbs are especially effective in highlighting the main practical takeaways.

Weaknesses

Scope / soundness limitations:
The theory is derived under a fairly specific setting with local latent variables refreshed via Gibbs updates. As acknowledged by the authors, this excludes more complex models with shared or interacting latent structures, such as hierarchical topic models or mixed-effects models. This limits the immediate generality of the results.

Practical robustness:
The main guarantees rely on a specific asymptotic polynomial scaling regime, especially the regime h + b = 1 for obtaining the most meaningful joint limit. It would strengthen the paper to discuss more explicitly how sensitive the method is in finite-sample settings when practical hyperparameter choices deviate from these idealized asymptotic rates.

Empirical scope / presentation:
While the experiments are well chosen, the empirical comparison is somewhat narrow. Most comparisons are against SVI; including stronger SG-MCMC baselines or more ablations on the role of S would strengthen the empirical case. In addition, a clearer runtime comparison between SGLD-Gibbs with different S values and SVI would help practitioners understand the computational trade-off behind the improved uncertainty calibration.

---

> ### Author Rebuttal · Authors · 2026-03-31
>
> We thank the reviewer for their thorough and constructive review. We are encouraged by their recognition of the originality of our joint scaling-limit analysis, the significance of translating asymptotic theory into concrete hyperparameter prescriptions, and the clarity of the paper's overall organization. We appreciate the constructive questions and suggestions, which we address below.
>
> Q1.
> Thank you for raising this point about finite-sample sensitivity. Since $h$ and $b$ are user-chosen, the condition $h+b=1$ can be explicitly targeted in practice rather than being a purely formal constraint. We focus on this regime because it yields a nondegenerate and interpretable joint limit; outside it, the latent-variable behavior changes qualitatively. For a precise quantification of finite-sample deviations, we refer the reviewer to Wang et al. (2025) ‘Quantitative Error Bounds for Scaling Limits of Stochastic Iterative Algorithms’, which provides a non-asymptotic functional approximation error bounds between the algorithm sample paths and the Ornstein–Uhlenbeck approximation in terms of sample size and hyperparameter. Note this is only for non-LVM cases.
>
> Q2.
> Our theory already disentangles these two effects. The tuning rules in Section 3.4 — choices of $h$, $\beta$, and the preconditioner — govern the scaling regime and target asymptotic covariance, independently of S. Increasing S under generalized Gibbs updates reduces only the latent-sampling contribution, changing the additional term from $M_\star$ to $M_\star/S$, without affecting the scaling regime itself since we assume $S$ is constant (independent of $n$)
>
> Q3.
> A key feature that leads to independence is that the global distribution of the latents is constant across iterations, hence they don't affect the global parameters. If the number of latent parameters $\to \infty$, we might expect similar behavior, but a careful analysis to account for the interactions between latent variables would be necessary. These interactions could, for example,  affect what relative scaling of the step size and batch size would be necessary. Extending the theory to such settings is an interesting direction but would require significant new technical machinery.
>
> Q4.
> This is a good point – we will include a clearer comparison in the camera-ready version.  Regarding computational cost, SVI converges in roughly 20 iterations in our synthetic GMM setting, while SGLD-Gibbs requires a burn-in phase of about 200 iterations plus additional iterations for posterior sampling. The per-iteration cost of SGLD is approximately $0.3-0.6\times$ that of SVI for $S=1$ and $1.6-3.0 \times$ for $S=5$ . And the total computational cost of SGLD-Gibbs depends on the number of desired posterior samples. SGLD-Gibbs cost between 5-10 times as much as SVI to converge, but the accuracy was improved by a factor of 20%-50% in different tasks and provided better uncertainty quantification. We will add details of computational costs in the camera-ready version.

---

### Decision · Program_Chairs · 2026-04-30

**Decision:**

Accept (regular)

**Comment:**

The authors present a paper on latent variable inference via a stochastic gradient Langevin dynamic Gibbs sampler that provides guidelines for hyperparameter tuning for the gradient-based sampler. This is a  novel contribution because these are hyperparameter settings that are generally tuned through rather brute methods, which hinders the overall time spent training SGLD-based samplers which are popular due to their scalability. The paper is well written, technically sound and theoretically justified. While the reviewers have raised the critique that the experiments are somewhat limited, I think the examples chosen are the classic latent variable models in Bayesian methods. The authors have suitably responded to the reviewers' concerns and I believe the contributions are substantial enough for publication at ICML.